# Unveiling Downstream Performance Scaling of LLMs: A Clustering-Based Perspective

**Chengyin Xu**,[*] **Kaiyuan Chen**,[*] **Xiao Li, Ke Shen, Chenggang Li**
Bytedance Seed
{xuchengyin.98, chenkaiyuan.99, lixiao.20}@bytedance.com
{shenke, lichenggang}@bytedance.com

## Abstract

The escalating scale and cost of Large Language Models (LLMs) training necessitate accurate pre-training prediction of downstream task performance for comprehensive understanding of scaling properties. This is challenged by: 1) the emergence phenomenon, where unpredictable capabilities appearing suddenly at critical model scales; and 2) uneven task difficulty and inconsistent performance scaling patterns, leading to high metric variability. Current prediction methods lack accuracy and reliability. We propose a Clustering-On-Difficulty (COD) framework for downstream performance prediction. The COD framework clusters tasks by their difficulty scaling features, thereby constructing a more stable and predictable task subset that exhibits well-behaved scaling characteristics with the increase of compute budget. We adopt a performance scaling law to predict cluster-wise performance with theoretical support. Predictable subset performance acts as an intermediate predictor for the full evaluation set. We further derive a mapping function to accurately extrapolate the performance of the subset to the full set. Applied to an LLM with 70B parameters, COD achieved a 1.55% average prediction error across eight key LLM benchmarks, thus providing actionable insights for scaling properties and training monitoring during LLM pre-training.

## 1 Introduction

Large Language Models (LLMs) have emerged as transformative technologies in natural language understanding, generation, and reasoning (Achiam et al., 2023; Guo et al., 2025; Bubeck et al., 2023). Their impressive success heavily relies on scaling model parameters and pre-training data, with training loss empirically following a power-law relationship with compute (Hoffmann et al., 2022; Kaplan et al., 2020). However, this reduction in training loss primarily reflects an in-domain compression effect and does not necessarily indicate improved out-of-domain generalization or downstream performance–the factor of primary concern in practice. Specifically, performance scaling of downstream tasks aims to predict the accuracy of the target LLM on downstream tasks using metrics from smaller models. Our objective is to develop a prediction method that works reliably on a diverse range of downstream tasks, optimizing the worst-case prediction error.

Despite extensive efforts, a reliable *scaling law for downstream tasks* remains elusive. One line of work attempts to extrapolate the performance of a large model by modeling the performance-loss relationship (Chen et al., 2024; Gadre et al., 2024; Du et al., 2024; Xiao et al., 2024; Owen, 2024), but this often fails to capture the emergent behaviors of LLMs and the mismatch between the in-domain loss and downstream metrics (Zhang et al., 2021). Another line of research focuses on direct extrapolation of the performance-compute relationship (Achiam et al., 2023; Hu et al., 2024), yet a single family of curves usually fails to capture the performance on evaluation benchmarks with complex difficulty distributions across samples.

A key limitation of existing methods is their unreasonable assumption that all evaluation samples follow a uniform performance scaling pattern. We observe that different evaluation samples actually follow distinct performance scaling patterns, and thus applying a single extrapolation formula to the entire evaluation set is suboptimal. We give a detailed analysis in Sec. 3.

---

[*]Equal contribution.

To address these challenges, we propose a new performance scaling law, derived from the existing loss scaling law (Kaplan et al., 2020), specifically applicable to evaluation subsets that exhibit consistent performance scaling patterns. Building on the performance scaling law, we develop a *Clustering-On-Difficulty* (COD) multi-stage framework for predicting downstream performance. Specifically, we first create a predictable subset by filtering out clusters that lack scaling properties using an improved MeanShift clustering algorithm. Next, we fit the performance-compute relationships in the predictable subset under our performance scaling law, extrapolate the performance of large models within each clusters, and finally map the aggregated predictions to the complete task set.

Crucially, the COD framework effectively resolves the challenges posed by emergent and heterogeneous behaviors. Regarding **non-emergent behaviors**, performance metrics for small models often fluctuate around random guessing or exhibit severe volatility, causing existing single-stage fitting methods to fail. Our method circumvents this by identifying a strong correlation between the predictable subset metrics and the full set metrics. This allows us to effectively estimate the full set performance using the predictable subset, where the relationship can be fitted with a smooth curve. Regarding **heterogeneous behaviors**, we observe that even within the predictable subset, different task clusters exhibit distinct scaling laws. By first performing cluster-wise extrapolation and then aggregating the results, COD can accurately capture the intrinsic heterogeneous scaling patterns within the evaluation set.

We validate our COD approach on eight popular evaluation sets, including MATH (Hendrycks et al., 2021), BBH (Suzgun et al., 2023), and MMLU pro (Wang et al., 2024) datasets. COD achieves an average prediction error of 1.55% on an LLM with 70B parameters. Our results demonstrate that this difficulty-aware framework substantially outperforms existing methods, establishing a promising paradigm for accurate downstream performance scaling of LLMs.

Our contributions can be summarized as follows.

- We propose the COD framework to address high variance and emergent phenomena in LLM performance scaling by effectively modeling the difficulty distribution within the evaluation sets.

- We introduce a downstream performance scaling law for cluster-wise performance prediction, with theoretical support and experimental validation.

- Extensive experiments conducted in eight different evaluation sets demonstrate that COD provides reliable predictions with an average prediction error of 1.55% on an LLM with 70B parameters.

## 2 RELATED WORK

### 2.1 LOSS SCALING LAWS

Loss scaling laws provide a systematic framework for understanding the relationship between computational resources, data, model size, and the LLM performance. Early work by Kaplan et al. (2020) demonstrates that the pre-training loss of LLMs follows a power-law relationship with the compute (the number of floating-point operations) used in training. Subsequent studies extend these findings to other domains, such as computer vision (Zhai et al., 2022), graph learning (Ma et al., 2024), and vision-language models (Alabdulmohsin et al., 2022; Henighan et al., 2020). Recent research has also explored scaling laws in specific contexts, such as fine-tuning (Hernandez et al., 2021; Tay et al., 2022), vocabulary size optimization (Tao et al., 2024), retrieval-augmented models (Shao et al., 2024), and hyperparameter tuning (Lingle, 2024; Yang et al., 2022). These studies highlight the broad applicability of scaling laws and their potential to guide the efficient allocation of computational resources.

### 2.2 DOWNSTREAM TASK PERFORMANCE SCALING

Predicting downstream task performance remains a critical challenge due to emergent abilities in LLMs that some capabilities manifest only after exceeding task-specific thresholds (Wei et al., 2022; Schaeffer et al., 2023). Recent works, such as using loss (Chen et al., 2024) or principal capability (Ruan et al., 2024) as a proxy, have demonstrated potential, but encounter challenges in aligning surrogate metrics with original task objectives. Other approaches manage to improve prediction accuracy by increasing the metric resolution (Hu et al., 2024) or incorporating experimental data from

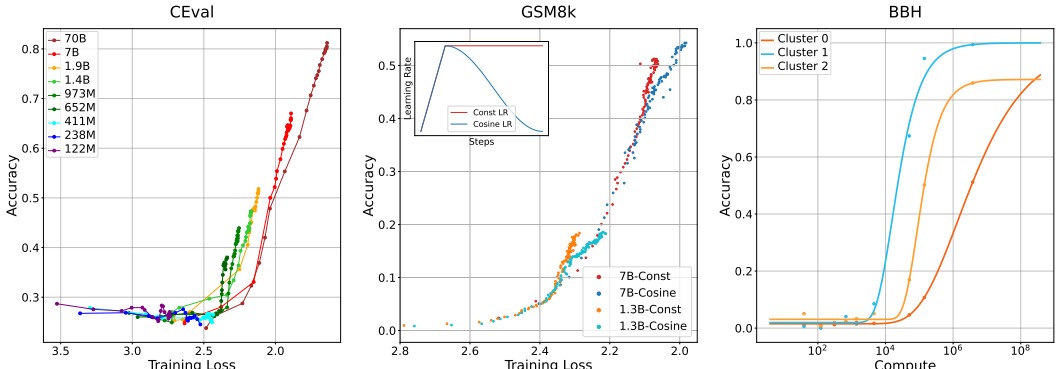

Figure 1: Performance-loss relationship across different model sizes (left) and learning rate schedules (middle). Performance-compute relationship for different clusters of the BBH samples(right)

other models (Ye et al., 2023). Here, we briefly review the two main types of methods for predicting downstream performance:

**Loss-intermediate prediction.** These methods predict the final training loss (or in-domain validation loss) of LLMs with loss scaling laws first, and then predict downstream performance through loss-performance relationships (Chen et al., 2024; Gadre et al., 2024; Du et al., 2024; Bhagia et al., 2024). While these methods leverage established scaling laws for loss predictions, they encounter a fundamental limitation: the inconsistent mapping between loss and performance metrics. In addition, Xiao et al. (2024) employ the evaluation set answer loss as an intermediate variable for estimation. Although answer loss correlates with the final performance metrics, its predictability remains low as predicting answer loss shares the challenges with predicting performance, including emergence phenomenon and high variance in task difficulty.

**End-to-end performance-compute prediction.** These methods (Hu et al., 2024; Owen, 2024; Achiam et al., 2023; Caballero et al., 2022) directly model the relationship between performance and the compute budget (or the number of model parameters). They are classified into exponential and piecewise types based on different formula formulations:

- Exponential methods: Achiam et al. (2023) estimate and fit this relationship using a subset of the evaluation set, while still failing to predict the full set. Hu et al. (2024) address the challenge of non-emergent capabilities in smaller models by employing multiple non-greedy decoding evaluations, thereby enabling accurate extrapolation of performance predictions for models with up to 2.4B parameters. However, it suffers from prohibitively high overhead during evaluation and can only predict non-greedy decoding metrics.

- Piecewise method: Caballero et al. (2022) propose a smooth broken power-law that models LLM scaling by decomposing it into multi-segment power laws. However, when predicting metrics for large-scale models (*e.g.*, 70B parameters), performance trends often exhibit unexpected inflection points due to emergent capabilities or saturation effects, making piecewise functions inadequate for capturing these novel scaling regimes.

## 3   PILOT STUDY

In this section, we present the pilot experiments to illustrate the shortcomings of existing approaches.

**Training loss may mismatch downstream task performance.** Predicting downstream performance from training loss assumes LLMs achieve identical downstream results at the same loss value, which does not hold universally. In practice, training loss primarily serves as an indicator of in-domain fitting, whereas downstream tasks typically represent out-of-domain evaluations. Moreover, training configurations, such as model size and learning rate, can significantly affect not only the final loss but also the model's generalization capabilities.

Fig. 1(left) illustrates the performance–loss relationships for LLMs of different sizes on the CEval benchmark (Seifert et al., 2024). At the same training loss level, smaller models can outperform larger ones in terms of test accuracy. Because smaller models initially exhibit weaker in-domain fitting capacity, they typically require more training steps to reach the same loss value, which can lead

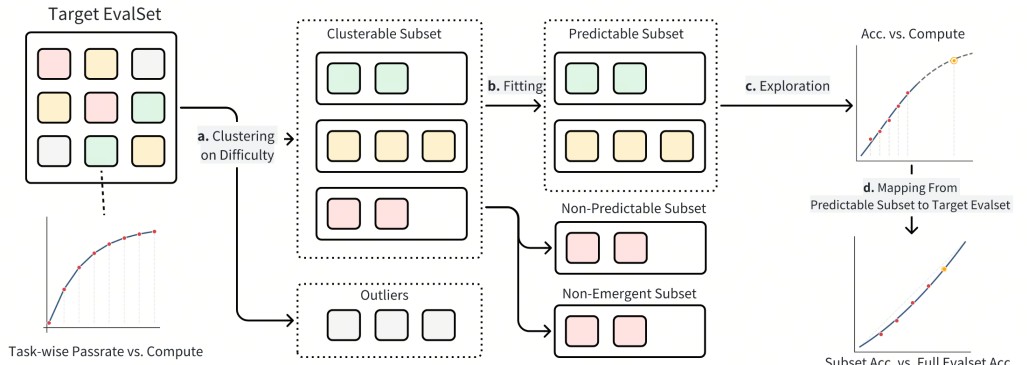

Figure 2: The pipeline of Cluster-On-Difficulty downstream task performance scaling, including 4 stages: **a**. Represent task difficulty feature with task-wise passrate vector. Cluster on the difficulty feature and filter outliers. **b**. Fit cluster-wise performance-compute curve. Classify clusters into extrapolatable clusters, non-extrapolatable clusters, and non-emergent clusters. **c**. Predict accuracy on extrapolatable clusters. **d**. Map subset accuracy prediction to full evaluation set performance.

to better in-domain generalization once they do. Fig. 1(middle) compares the performance of LLMs trained under different learning rate schedules on the GSM8k dataset (Cobbe et al., 2021). At the same loss level, the performance under the cosine schedule is always worse than that under the constant schedule, indicating that a lower learning rate may prioritize memorization over generalization, thereby diminishing downstream performance.

**Diverse scaling patterns within the evaluation set.** Different task samples exhibit unique computational thresholds, learning slopes, and upper bounds, making it challenging to find a single fitting function (or function group) that generalizes well across diverse task samples. Fig. 1(right) illustrates the performance-compute relationships on three clusters randomly selected from those formed by clustering tasks based on their difficulty in the BBH benchmark. (Suzgun et al., 2023), with each cluster containing samples with similar difficulty. Even within a single evaluation set, these scaling curves can vary significantly, indicating that a one-size-fits-all performance-compute curve is insufficient for capturing the full spectrum of a downstream evaluation set.

Taken together, these observations highlight the importance of modeling the heterogeneous scaling properties within an evaluation set and identifying a robust intermediate metric to serve as a reliable indicator of the downstream performance of LLMs.

## 4 METHOD

In this section, we first formulate the problem, then present COD in four stages (see Fig. 2). (1) We construct sample-level difficulty scaling features and apply an improved MeanShift clustering algorithm (Sec. 4.1). (2) We derive a performance scaling law with respect to task difficulty variance, enabling extrapolation of performance–compute relationships for clusters with similar difficulty features. Cluster-wise curves are fitted on small models to identify extrapolatable clusters (Sec. 4.2). (3) We extrapolate performance for these clusters to predict the target large model's accuracy on the predictable subset (Sec. 4.3). (4) Finally, we map subset accuracy to full evaluation results (Sec. 4.4).

**Problem Formulation.** Consider a language model $M_C$ trained with a compute budget of $C$ measured in FLOPs. Let $\mathcal{P}$ be a set of downstream tasks that we aim to evaluate the model on. Each sample $T \in \mathcal{P}$ is defined by a question-answer pair $(q, a_{\text{true}})$. Given a question $q$, the model $M_C$ outputs a probability distribution $p(a|q; M_C)$ over the space of all possible answers.

Our goal is to predict the downstream task performance of a large language model $M_{C_{\text{target}}}$ using only evaluation results from smaller models $\{M_{C_1}, M_{C_2}, \dots, M_{C_n}\}$ where $C_i \ll C_{\text{target}}$ for all $i$. Formally, we aim to find the prediction method $\phi$ to minimize the absolute prediction error over a

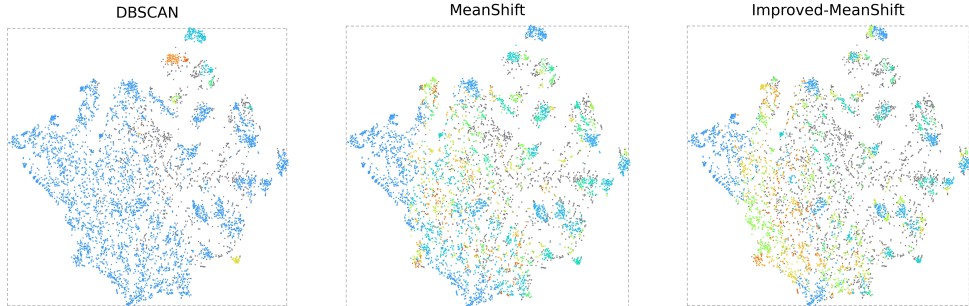

Figure 3: t-SNE visualization of different clustering methods: DBSCAN(left), MeanShift(Middle), Improved-MeanShift(Right). Each point represents an evaluation sample.

group of tasks sets $\{\mathcal{P}_j\}_m$:

$$\arg\min_{\phi} \frac{1}{m} \sum_{i=1}^{m} \frac{1}{|\mathcal{P}_j|} \sum_{T \in \mathcal{P}_j} |\widehat{\mathrm{Acc}}(C_{\mathrm{target}}, T) - \mathrm{Acc}(C_{\mathrm{target}})|,$$

$$\widehat{\mathrm{Acc}}(C_{\mathrm{target}}, T) := \phi\left(\{\mathrm{Acc}(C_i, T)\}_{i=1}^{n}, \{C_i\}_{i=1}^{n}, C_{\mathrm{target}}\right),$$

where $\mathrm{Acc}(C, T)$ denotes the accuracy of model $M_C$ on task $T$, and $\widehat{\mathrm{Acc}}(C_{\mathrm{target}}, T)$ is the predicted accuracy for the target model.

## 4.1 CLUSTERING ON DIFFICULTY

Although downstream tasks in the same evaluation set share similar themes, they exhibit significant differences in difficulty, resulting in distinct performance scaling patterns that make a universal fitting function inapplicable. We propose clustering tasks by similar performance scaling behaviors to minimize intra-cluster heterogeneity maintaining a minimum cluster size.

Specifically, we train a group of language models with increasing parameter counts. These models are trained with the same ratio of training tokens to compute per token. We use the same set of small models for prediction to evaluate the difficulty characteristics of the task, and will not introduce the target large model evaluation results to avoid feature leakage.

For each task, we generate 100 samples using `top_p`=0.7 and `temperature`=1.0 for each model, and compute the pass rate by averaging the results. This pass rate serves as an estimate of the model's expected accuracy on the task. The resulting values are concatenated into a difficulty vector, ordered by increasing model size. For most tasks, this difficulty vector exhibits a monotonic increase, reflecting the gradual improvement of model capability with scale.

After obtaining the difficulty feature vector for each task, we use the improved clustering algorithm that incorporates the following features: (1) Minimizing intra-class variance to ensure similar extrapolation properties within each cluster; (2) Automatic determination of cluster numbers, as the optimal number varies across evaluation sets and is difficult to pre-specify.

To further reduce intra-class variance, we propose an improved MeanShift algorithm to constrain the cluster diameter. At the same time, we maintain a minimum number of tasks in each cluster to reduce metric fluctuations. We provide the t-SNE visualization of clustering results evaluation tasks on BBH (Suzgun et al., 2023) to compare the proposed method and classic clustering algorithms including DBSCAN (Ester et al., 1996) and MeanShift (Fukunaga and Hostetler, 1975). Each point represents an evaluation sample, and its color denotes the cluster type. As shown in Fig. 3, our improved MeanShift effectively splits dense areas whereas DBSCAN and the original MeanShift produce connected clusters with large within-cluster distances.

We provide numerical comparison of clustering algorithms in Sec. 5.3.2 and explain implementation details of improved MeanShift in Appendix A.1, smoothing techniques in Appendix A.2.

## 4.2 FITTING

After clustering, we compute metrics for small models within each cluster. We then fit the accuracy-compute curves for each cluster using a theoretically derived novel performance scaling law, focusing on clustered samples after excluding outliers. We derive the following fitting formula for the downstream task scaling law based on the training loss scaling law.

**Theorem 1** (Scaling Law for Downstream Task Performance). *Consider a language model $M_C$ trained with compute budget $C$ and a set of downstream tasks $\mathcal{P}$. Under the following assumptions: Assumption 1 (Power-law scaling of answer loss): the expected answer loss follows:*

$$L_{\mathcal{P}}(C) := \mathbb{E}_{(q,a_{true})\sim\mathcal{P}}[L(q, a_{true}; C)] = \alpha C^{-\beta} + \gamma, \tag{1}$$

*where $\alpha, \beta, \gamma > 0$ are task-specific constants, with $\gamma$ representing the irreducible loss.*

*Assumption 2 (Unique deterministic answers): Each question has a unique deterministic answer. The model receives score $1$ if and only if $M_C$ outputs $a_{true}$, and $0$ otherwise.*

*Assumption 3 (Accuracy decomposition): The expected accuracy decomposes as:*

$$\mathbb{E}_{T\sim\mathcal{P}}[\text{Acc}(C)] = g + (1 - g) \cdot \mathbb{E}_{(q,a_{true})\sim\mathcal{P}}[p(a_{true}|q, M_C)], \tag{2}$$

*where $g \in [0, 1]$ is the random guessing baseline.*

*Then, the expected accuracy on task set $\mathcal{P}$ can be modeled as:*

$$\mathbb{E}_{\mathcal{P}}[\text{Acc}(C)] = g + (1 - g)\left(\exp\left(-\alpha C^{-\beta} - \gamma\right) + \frac{\sigma_L^2(C)}{2\mu_L(C)}\right) + o\left(\sigma_L^2(C)\right), \tag{3}$$

*where $\mu_L(C) = \mathbb{E}_{(q,a_{true})\sim\mathcal{P}}[L(q, a_{true}; C)]$ is the mean loss and $\sigma_L^2(C) = \text{Var}_{(q,a_{true})\sim\mathcal{P}}[L(q, a_{true}; C)]$ is the loss variance across the task set.*

*Proof Sketch.* By definition of the language model loss, $p(a_{\text{true}}|q, M_C) = \exp(-L(q, a_{\text{true}}; C))$. Under Assumption 1, if the answer loss follows a power law $L \sim \alpha C^{-\beta} + \gamma$, then the task passrate should approximately scale as $\exp(-\alpha C^{-\beta} - \gamma)$.

The key subtlety lies in the averaging: accuracy computes $\mathbb{E}[\exp(-L)]$ (arithmetic mean of passrates) while the loss scaling law gives us $\exp(-\mathbb{E}[L])$ (geometric mean). Using Taylor expansion:

$$\mathbb{E}[\exp(-L)] \approx \exp(-\mu_L) + \frac{\sigma_L^2}{2\mu_L},$$

where $\mu_L$ and $\sigma_L^2$ are the mean and variance of the loss distribution.

This approximation is accurate when tasks have similar difficulty feature ($\sigma_L^2/\mu_L^2 \ll 1$), motivating our clustering approach to reduce intra-cluster variance. Assumption 3 adds the parameter $g$ for random guessing. The complete proofs are provided in Appendix B. $\square$

Theorem 1 demonstrates that a metric of an evaluation set with similar difficulty features can be effectively modeled using the following formula:

$$y(C) = g + (1 - g) * e^{-aC^{-b} - c}, \tag{4}$$

where $a$ and $b$ jointly influence how accuracy varies with $C$, while $c$ constrains the upper bound of the fitting curve, and $g$ represents the expected random guess metric for a task cluster. $a$, $b$, $c$, and $g$ are trainable parameters. Note that these assumptions may not perfectly hold in practice, we provide additional discussions on assumption 3 in Appendix H.

## 4.3 EXTRAPOLATION

To ensure reliable extrapolation, we identify clusters exhibiting robust scaling patterns, as some clusters may show saturated or non-emergent performance on smaller models, making them unsuitable for prediction. We aim to find an extrapolation subsets that represent the full set performance, and use the subset metric as a intermediate indicator for the prediction of the full set accuracy.

A cluster is deemed **extrapolatable** if it meets two criteria: (1) its expected accuracy increases monotonically with model size, and (2) its performance converges to at least a predefined threshold $P$ (where $P \leq 1$ accounts for practical limits like ambiguous questions or finite training coverage).

We filter out non-extrapolatable clusters using two rules based on the parameters from Eq. (4):

1. Negligible accuracy growth, indicated by minimal $a$ or $b$ values.

2. Poor extrapolation reliability, indicated by an excessive $c$ value.

In practice, for extrapolatable clusters, we set $a > 1$, $b > 0.1$, and $0 \leq c < 1$. Further ablation experiments are provided in Appendix C.1.

The clusters satisfying these conditions form the *predictable subset*. The final performance prediction for a target model on this subset is the weighted average of the extrapolated predictions from these individual clusters, with weights proportional to cluster sizes.

### 4.4 MAPPING FROM PREDICTABLE SUBSET TO TARGET EVALUATION SET

We map predictions from the predictable subset $\mathcal{P}'$ to the complete evaluation set $\mathcal{P}$ using a smooth function. This mapping strategy is motivated by the observation that extrapolatable and non-extrapolatable samples, despite their difficulty differences, usually belong to the same question types, which implies a consistent relative metric ordering between the predictable subset and the full evaluation set. The mapping function $f : \text{Acc}(\mathcal{P}') \to \text{Acc}(\mathcal{P})$ is continuous, smooth over $[0, 1]$, monotonically increasing, and constrained to pass through $(0, 0)$ and $(1, 1)$. Empirical validation indicates that a smoothing spline optimally captures this relationship. Specifically, we employ a cubic smoothing spline to model the mapping. where $x$ represents the average accuracy of the predictable subset $\mathcal{P}'$. In practical implementation, under the premise of fixing the curve to pass through $[0, 0]$ and $[1, 1]$, we determine the number of piecewise cubic segments (knots) by setting a Root Mean Square Error (RMSE) fitting threshold of $0.005$. The number of segments is dynamically adjusted until the fitting RMSE meets this threshold. We list the implementation details and visualization of mapping in Sec. C.2.

To ensure reliability, we calibrate $f$ using evaluation results from existing models as anchors. This subset-to-full mapping generally demonstrates robustness across diverse model architectures and training data, often permitting the use of external models (e.g., Qwen2-72B (Yang et al., 2024b)) as anchors for many tasks (see Appendix C.3 for experiments). The final metric prediction for a target LLM with estimated training computation $C_0$ is then $p = f \circ y(C_0)$, combining the cluster-wise extrapolation $y(C_0)$ from Eq. (4) with the mapping $f$.

## 5 EXPERIMENTS

### 5.1 EXPERIMENTAL SETUPS

In our experimental setup, we train nine language models ranging from 122M to 70B parameters in total, which share the same data distribution and architecture, with the training data scaled proportionally to their sizes. We show the detailed training configurations and recipe in Appendix D.

For evaluation, we adopt the following widely used benchmarks, including GSM8K (Cobbe et al., 2021), MATH (Hendrycks et al., 2021), BBH (Suzgun et al., 2023), TriviaQA (Joshi et al., 2017), MBPP (Austin et al., 2021), AGIEval (Zhong et al., 2024), DROP (Dua et al., 2019), MMLU-pro (Wang et al., 2024). All models are evaluated in a few-shot in-context learning manner, and we aligned our evaluation setups with LLaMa3 (Dubey et al., 2024).We evaluate the proposed COD performance scaling for LLMs against existing approaches on multiple public benchmarks. Using eight smaller language models as known information, we estimate the downstream task performance of a pretrained LLM with 70B parameters.

### 5.2 PREDICTION EXPERIMENTS

We compare COD against four representative prediction methods:

Table 1: Absolute prediction error (%) on evaluation sets for predicting the performance of the 70B model. Errors < 2% are considered accurate (green), while errors > 5% are considered invalid (red). ↓ indicates lower is better.

| Method | Overall Metrics | | Individual Task Sets | | | | | | | |
|---|---|---|---|---|---|---|---|---|---|---|
| | Mean↓ | Max↓ | GSM8k | MATH | BBH | TriviaQA | MBPP | AGIEval | DROP | MMLU-pro |
| Loss-intermediate | 5.29 | 9.39 | 9.39 | 6.95 | 2.33 | 5.81 | 5.52 | 1.41 | 5.37 | 5.55 |
| End-to-end(exp) | 3.10 | 6.00 | 4.00 | 3.86 | 0.64 | 0.68 | 1.75 | 6.00 | 4.11 | 3.72 |
| End-to-end(passrate) | 5.02 | 8.80 | 6.71 | 8.80 | 3.51 | 4.00 | 7.34 | 6.78 | 0.26 | 2.74 |
| End-to-end(BNSL) | 5.17 | 13.05 | 4.23 | 5.88 | 13.05 | 5.86 | 2.55 | 0.82 | 1.53 | 7.42 |
| COD (w/o mapping) | 2.24 | 5.26 | 4.70 | 0.50 | 2.91 | 1.98 | 0.89 | 5.26 | 1.08 | 0.57 |
| COD (Complete) | **1.55** | **2.68** | 2.68 | 0.79 | 0.47 | 1.97 | 2.42 | 1.64 | 1.05 | 1.39 |

1. Loss-intermediate (Chen et al., 2024): First predicts the target LLM's final training or validation loss, then estimates downstream task metrics based on the relationship between smaller models' evaluation metrics and their losses.

2. End-to-end(exp) (Xiao et al., 2024): Directly extrapolates large model metrics from smaller model evaluation set metrics using exponential-based performance scaling laws.

3. End-to-end(passrate) (Achiam et al., 2023; Hu et al., 2024): A variant of end-to-end method, which estimates large model passrates from smaller model passrates. We conduct 100 trials per evaluation set for smaller models to enhance reliability and report absolute prediction error on the passrate metric.

4. End-to-end(BNSL) (Caballero et al., 2022): Decomposes the end-to-end mapping into a multi-segment power-law framework.

We also evaluate two variants of our COD approach to validate the benefits of its components:

1. COD (w/o mapping): Performs difficulty-based KMeans clustering, extrapolates per cluster, and aggregates metrics without subset-to-full mapping.

2. COD (Complete): Our full proposed multi-stage approach, including clustering, predictable cluster filtering, subset extrapolation, and subset-to-full mapping.

Comparative results are shown in Tab. 1. Prediction accuracy is measured by the absolute error between predicted and actual performance. We report mean and max prediction errors across all evaluation sets, as well as errors for individual sets. Our complete COD approach significantly outperforms existing methods in both mean (1.55%) and maximum (2.68%) prediction errors, offering reliable guidance for large model training. Although baseline methods achieve acceptable performance on partial datasets, their large prediction errors on other datasets severely compromise their overall reliability.

Fig. 4 visualizes the performance-compute relationships. The COD method does not merely extend the existing scaling trend; instead, it effectively predicts whether growth slowdown will occur subsequently and enables better estimation of the magnitude of curve bending. On the BBH dataset, while End-to-end(exp) and loss-intermediate approaches perform comparably to COD, they show poor fitting on small-model data. COD reveals a more complex, better-fitted multi-phase trajectory. On MATH and MMLU-pro, where predicting accelerated growth versus plateaus is crucial, the loss-intermediate method underestimates model ceilings, and two end-to-end methods exceeds 3% error. COD's superior performance stems from its nuanced analysis of difficulty distributions and scaling laws, allowing it to predict growth in challenging sets and capture diminishing returns in saturated sets.

## 5.3 ABLATION STUDY

To further validate the generalizability of the COD method, we conduct ablation experiments on different architecture, clustering method and extrapolation function, while in the Appendix C we discuss ablation studies on the selection criteria for predictable subsets, interpolation mapping methods, and the influence of anchor point on predictions. We also examine its predictive capabilities after continual training with results documented in Appendix E.

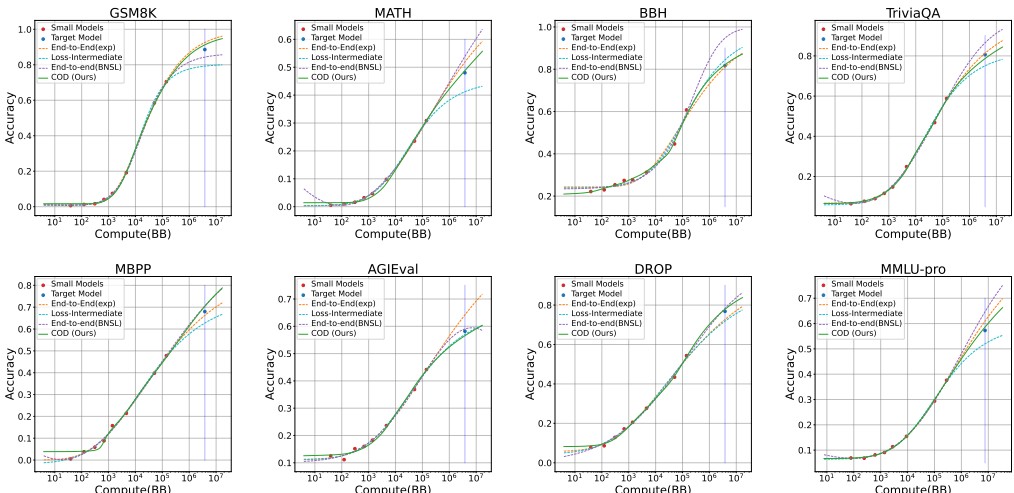

Figure 4: Performance-compute relationship for different prediction methods on eight evaluation sets.

Table 2: Absolute prediction error (%) on evaluation sets for predicting the performance of the 32B MoE model. Errors < 2% are considered accurate (green), while errors > 5% are considered invalid (red). ↓ indicates lower is better.

| Method | Overall Metrics | | Individual Task Sets | | | | | | |
|---|---|---|---|---|---|---|---|---|---|
| | Mean↓ | Max↓ | GSM8k | MATH | BBH | TriviaQA | MBPP | AGIEval | DROP |
| Loss-intermediate | 3.65 | **7.24** | 0.45 | 0.62 | 4.48 | 0.36 | 4.92 | 7.24 | 4.55 |
| End-to-end(exp) | 3.95 | 7.86 | 2.75 | 1.43 | 3.88 | 1.72 | 7.86 | 7.79 | 2.21 |
| COD (Complete) | **3.11** | 8.11 | 0.54 | 1.72 | 5.29 | 0.27 | 8.11 | 0.57 | 5.24 |

### 5.3.1 PREDICTION ON MOE MODELS

COD relies on repeated sample-level evaluation and clustering, introducing additional computational overhead. However, difficulty characteristics are task-inherent and largely model-agnostic, suggesting that the resulting clusters can generalize across model families.

To test this transferability, we conduct one-shot evaluations on a 32B activated-parameter MoE model using clusters derived from pre-trained dense models. Results in Tab. 2 show that COD achieves the lowest average prediction error in cross-architecture extrapolation, indicating that its difficulty features and clusters transfer effectively across model families. Nevertheless, prediction accuracy is lower than in dense-to-dense extrapolation. We hypothesize that aligning the model used for difficulty estimation with the target model for extrapolation reduces intra-cluster scaling discrepancies and improves prediction accuracy.

### 5.3.2 COMPARISON OF CLUSTERING METHODS

We assess the impact of different clustering algorithms on prediction accuracy. The goal is to achieve tight intra-cluster difficulty similarity (low average distance to center) while maintaining cluster stability (min. 10 samples/cluster). We compare our Improved-MeanShift with DBScan, MeanShift, and K-Means. For K-Means, we adjust it to approximate our goals: (1) search for $k$ yielding min. cluster sizes $\approx 10$; (2) treat samples outside a radius (e.g., $2\times$ average intra-cluster distance) from any cluster center as outliers, and we ensure clusters don't drop below 10 samples. We term this "Improved-KMeans" for this comparison. Clustering quality is measured by Intra-cluster Average Distance (IAD) and Outlier Rate (OR). Prediction benefits are measured by Extrapolation Error (EE) on the predictable subset and Final prediction Error (FE) on the full evalset. (Tab. 3).

Tab. 3 shows Improved-KMeans and Improved-MeanShift yield better clustering (lower IAD) due to their intra-cluster distance constraints. The results also confirm these methods lead to lower EE and FE. Although Improved-KMeans has the best IAD, it performs poorly on GSM8k, AGIEval, and

Table 3: Clustering performance across different algorithms on metrics including IAD (Intra-cluster Average Distance↓), OR (Outlier Rate %), EE (Extrapolation Error↓), FE (Final Prediction Error↓). The bottom lines show the mean and max EE and FE across evalsets.

| Dataset | KMeans | | | | DBScan | | | | MeanShift | | | | Improved-KMeans | | | | Improved-MeanShift | | | |
|---|---|---|---|---|---|---|---|---|---|---|---|---|---|---|---|---|---|---|---|---|
| | IAD | OR | EE | FE | IAD | OR | EE | FE | IAD | OR | EE | FE | IAD | OR | EE | FE | IAD | OR | EE | FE |
| GSM8k | 0.22 | - | **0.01** | **0.00** | 0.51 | 0.53 | 4.08 | 4.12 | 0.29 | 0.61 | 0.67 | 0.74 | **0.13** | 2.73 | 3.92 | 4.08 | 0.16 | 7.05 | 0.31 | 2.68 |
| MATH | 0.22 | - | 2.62 | 2.34 | 0.48 | 0.68 | 4.38 | 4.16 | 0.21 | 1.44 | 2.55 | 2.26 | **0.09** | 2.22 | **0.81** | **0.51** | 0.11 | 6.26 | 0.84 | 0.79 |
| BBH | 0.63 | - | 8.16 | 8.99 | 0.71 | 18.92 | 3.53 | 4.36 | 0.27 | 20.72 | 2.12 | 0.65 | **0.20** | 37.23 | **0.02** | 2.17 | 0.21 | 33.58 | 0.54 | **0.47** |
| TriviaQA | 0.44 | - | 2.97 | 2.46 | 0.70 | 6.38 | **1.11** | **0.81** | 0.25 | 6.77 | 3.64 | 4.90 | **0.12** | 11.97 | 1.18 | 1.12 | 0.19 | 11.54 | 1.56 | 1.97 |
| MBPP | 0.34 | - | 2.53 | 2.67 | 0.51 | 12.80 | **1.57** | 1.41 | 0.22 | 15.60 | 2.40 | **1.22** | **0.17** | 19.40 | 2.39 | 3.25 | **0.17** | 21.60 | 1.61 | 2.42 |
| AGIEval | 0.46 | - | 2.61 | 2.68 | 0.56 | 3.67 | 6.43 | 6.27 | 0.29 | 2.99 | 2.63 | 3.23 | **0.15** | 7.60 | 5.96 | 5.56 | 0.21 | 11.50 | **1.11** | **1.64** |
| DROP | 0.56 | - | 1.66 | 1.64 | 0.67 | 11.08 | 3.03 | 2.66 | 0.25 | 11.81 | 4.18 | 4.00 | **0.14** | 21.42 | 3.99 | 5.24 | 0.20 | 19.88 | **1.44** | **1.05** |
| MMLU-pro | 0.32 | - | 3.69 | 3.69 | 0.42 | 0.56 | 3.72 | 3.69 | 0.29 | 0.39 | 3.15 | 3.08 | **0.16** | 2.85 | **0.56** | **0.61** | 0.22 | 4.40 | 1.26 | 1.39 |
| **Mean** | - | - | 3.03 | 3.06 | - | - | 3.48 | 6.43 | - | - | 2.67 | 2.51 | - | - | 2.35 | 2.82 | - | - | **1.08** | **1.55** |
| **Max** | - | - | 8.16 | 8.99 | - | - | 4.38 | 4.36 | - | - | 4.18 | 4.90 | - | - | 5.96 | 5.56 | - | - | **1.61** | **2.68** |

Table 4: Ablation study on extrapolation formulas. EE, TR, FE shown for BBH, MATH, MMLU-pro.

| Method | BBH | | | MATH | | | MMLU-pro | | |
|---|---|---|---|---|---|---|---|---|---|
| | EE↓ | TR(%) | FE↓ | EE↓ | TR(%) | FE↓ | EE↓ | TR(%) | FE↓ |
| w/o Random Guess ($f_1$) | 10.40 | 48.29 | 11.65 | 3.96 | 76.82 | 3.22 | 4.40 | 95.05 | 4.32 |
| w/o Constant c ($f_2$) | 2.15 | 57.21 | 4.10 | 1.50 | 76.82 | 1.36 | 3.85 | 95.60 | 3.96 |
| Direct Power Law ($f_3$) | 8.90 | 49.05 | 8.85 | 3.33 | 76.82 | 2.70 | 4.30 | 95.15 | 4.20 |
| Ours ($f$) | **0.54** | 53.39 | **0.47** | **0.84** | 76.82 | **0.79** | **1.26** | 94.38 | **1.39** |

DROP. This is likely because K-Means requires pre-specifying $k$, and our search for $k$ can be unstable, leading to large errors on some sets. In contrast, our Improved-MeanShift, which automatically determines $k$ based on distance constraints, offers more stable clustering and the lowest maximum prediction error.

### 5.3.3 DIFFERENT EXTRAPOLATION FORMULAS

We ablate our proposed fitting formula $f(C) = g + (1 - g) \cdot e^{-aC^{-b}-c}$ (Ours) by removing or modifying components: (1) $f_1(C) = e^{-aC^{-b}-c}$ (w/o random guess); (2) $f_2(C) = g + (1 - g) \cdot e^{-aC^{-b}}$ (w/o constant $c$); (3) $f_3(C) = e^{-aC^{-b}}$ (Direct Power Law (Hu et al., 2024)). Tab. 4 shows Extrapolation Error (EE), Task Ratio of predictable subset (TR), and Final prediction Error (FE). Our proposed formula $f$ consistently achieves the lowest EE and FE. $f_1$ struggles with finite-answer tasks where small models have non-zero scores. $f_2$ inaccurately assumes perfect scores are attainable, ignoring data limitations and task ambiguities. The direct power law ($f_3$) fails to model the 0-1 metric range and the varying difficulty of improvement near random guess and saturation. The weak correlation between TR and prediction error demonstrates the robustness of our COD framework: even when the proportion of the predictable subset is low due to non-emergent tasks, the performance of non-extrapolatable tasks can still be accurately inferred through the proposed mapping function.

## 6 CONCLUSION AND DISCUSSION

In this work, we propose a novel framework for predicting LLM downstream performance scaling, which consists of three key contributions: (1) the COD framework that effectively models the intrinsic diverse scaling patterns of tasks in the evaluation set; (2) a scaling law for downstream task performance that provides a fitting formula for performance-compute extrapolation; and (3) a systematic methodology for identifying and leveraging predictable subset that provides a robust intermediate metric for accurate full-set performance predictions. We discuss the limitations and future works in Appendix H.

ETHICS STATEMENT

We have read and adhered to the ICLR Code of Ethics. This work proposes a computational framework to enable more efficient resource allocation in the training of LLMs. The research is methodological in nature and aims to support more sustainable and responsible practices within the field of AI development.

We provide a detailed account of our methods, theoretical proofs, and experimental settings. We openly discuss the limitations of our framework in Appendix H. This study does not involve human subjects or the use of sensitive personal data.

We utilized language models at the writing level, including checking for grammatical errors in the article and modifying expressions. The use of language models had no impact on the article's innovative contributions, experiments, or analytical perspectives.

REPRODUCIBILITY STATEMENT

We have made extensive efforts to ensure the reproducibility of our work. The core methodology, the Clustering-On-Difficulty (COD) framework, is detailed in Sec. 4. The improved MeanShift clustering algorithm is described in Sec. 4.1 , with full pseudocode provided in Appendix A.1 (Algorithm 1). Our performance scaling law (Theorem 1) is presented in Sec. 4.2 , with a complete proof available in Sec. B. We discuss the additional computational cost of the COD method in Appendix G.

Our experimental setup, including model architectures, training data philosophy, and hyperparameters, is thoroughly documented in Sec. 5.1 and Appendix D , with specific model configurations listed in Tab. A4. The evaluation benchmarks, protocols, and few-shot settings are described in Sec. 5.1 and summarized in Tab. A5. Extensive ablation studies validating our component choices—including extrapolation formulas (Sec. 5.3.3), clustering algorithms (Appendix 5.3.2), interpolation methods (Appendix C.2), and criteria for filtering clusters (Appendix C.1)—are provided to support our findings. We visualize the task difficulty distribution for each evalset in Appendix F.

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

CONTENTS

# A  IMPROVEMENTS OF CLUSTERING ALGORITHM

## A.1  IMPROVED MEANSHIFT ALGORITHM

We iteratively apply the MeanShift algorithm with an adaptive cluster radius $R$ and a minimum cluster size $K$. In each iteration, for the clustered samples, we examine whether the distance between each sample and its cluster center exceeds R, and relabel those samples that exceed this threshold as unclustered. For clusters containing fewer than K samples, we mark all samples in these clusters as unclustered. At the end of each iteration, we incorporate both the outliers from MeanShift and our marked unclustered samples into the next round of clustering, continuing this process until no further changes occur in sample labels. We present the pseudocode in Algorithm 1.

---

**Algorithm 1** Iterative MeanShift Clustering Algorithm

---

1: Calculate adaptive radius: $R = \min(\text{estimate\_bandwidth}(Q), U)$
2: Initialize all labels in the evaluation set to $-1$
3: **repeat**
4:     Perform MeanShift clustering with radius $R$ on all samples labeled $-1$
5:     Assign new labels to clustered samples
6:     **for** each newly labeled sample $i$ **do**
7:         Calculate distance $\text{dist}_i$ to its cluster center
8:         **if** $\text{dist}_i > R$ **then**
9:             Reset label to $-1$
10:         **end if**
11:     **end for**
12:     **for** each cluster **do**
13:         **if** number of samples in cluster $< K$ **then**
14:             Reset all samples in this cluster to $-1$
15:         **end if**
16:     **end for**
17:     Renumber all non-$\{-1\}$ newly labeled samples to avoid overlap with old labels
18: **until** no label changes

---

In the experiment, $K$ is empirically set to $10$, which has been verified through extensive experiments to be a reasonable and robust value. To determine the clustering radius $R$, we employ the *estimate_bandwidth* function from the *sklearn.cluster* library. This utility automatically computes a bandwidth value that balances clustering granularity and stability based on the underlying distribution of the data. The function is governed by a quantile hyperparameter $Q$, which typically ranges from $0.1$ to $0.5$. Given our framework's stringent requirements for minimizing intra-cluster variance, we adopt a conservative value of $Q = 0.1$ in practice. Furthermore, considering that sample sizes and difficulty distributions vary significantly across different evaluation sets, the automatically estimated bandwidth may occasionally become excessively large. To mitigate this risk, we introduce a global upper bound $U$ on the bandwidth. Specifically, we define $U = max\_distance/10$, where max_distance represents the theoretical maximum distance between any two vectors in the difficulty feature space and 10 is a empirical constant. In our experimental setup, the feature vectors are 9-dimensional with each element inherently bounded within the range $[0, 1]$. Consequently, the maximum possible Euclidean distance is calculated as $\sqrt{1^2 \times 9} = 3$, yielding an upper bound of $U = 0.3$.

**Filtering zero-performance samples.**  In the evaluation set, there may exist a few extremely difficult problems that require sufficient model parameters to emerge. All small models may fail to solve these problems even after 100 evaluation attempts, resulting in difficulty feature vectors of all zeros. We refer to these as zero-performance samples. Their presence leads to two issues:

1. Zero performance on small models does not necessarily indicate zero accuracy on large models. For these samples, we cannot estimate when emergence will occur or predict large model metrics.

2. During clustering, they may be confused with other low-performing but non-zero samples. Including them in the same cluster would lower the expected accuracy of that cluster, leading to inaccurate fitting and extrapolation later.

Therefore, we pre-filter these zero-performance samples before clustering, treating them as outliers that do not participate in the clustering process. This approach obviates the necessity of considering their metrics under large models during subsequent extrapolation, and prevents disruption to the clustering of the remaining samples.

## A.2 Smoothing Techniques

Metric fluctuations of individual samples in downstream tasks are not solely due to limited sampling. Another potential factor is noise from uneven data distribution in recent training batches. Therefore, in addition to performing 100 evaluations to mitigate sampling variance, we evaluated 100 times on each of the adjacent checkpoints before and after the selected model. We then averaged these accuracy expectation values across three checkpoints, further reducing sampling variance while offsetting noise from uneven training data distribution. This approach also reduces the number of zero-performance samples, further improving clustering and prediction effectiveness.

## B  Proof of Theorem 1

We use Lemma 1 to derive the scaling law for downstream task performance (Theorem 1).

**Lemma 1** (Arithmetic-geometric mean difference). *For any sequence of positive real numbers* $\{x_i\}_{i=1}^n$, *let:*

- $\mu_a = \frac{1}{n}\sum_{i=1}^n x_i$ *be the arithmetic mean;*

- $\mu_g = \prod_{i=1}^n x_i^{1/n}$ *be the geometric mean;*

- $\sigma^2 = \frac{1}{n}\sum_{i=1}^n (x_i - \mu)^2$ *be the variance.*

*Then the difference between the arithmetic mean and geometric mean can be estimated as:*

$$\Delta = \mu_a - \mu_g = \frac{1}{n}\sum_{i=1}^n x_i - \left(\prod_{i=1}^n x_i\right)^{\frac{1}{n}} = \frac{\sigma^2}{2\mu_a} + o(\sigma^2). \tag{5}$$

*Proof.* Taking the logarithm of the geometric mean $\mu_g$:

$$\log(\mu_g) = \frac{1}{n}\sum_{i=1}^n \log x_i. \tag{6}$$

Using Taylor expansion of $\log x$ around $\mu$:

$$\log x = \log \mu + \frac{x - \mu}{\mu} - \frac{(x-\mu)^2}{2\mu^2} + o\left((x-\mu)^2\right) \tag{7}$$

We can simplify:

$$
\begin{aligned}
\frac{1}{n}\sum_{i=1}^n \log x_i &= \log \mu + \frac{1}{n}\sum_{i=1}^n \left(\frac{(x_i - \mu_a)}{\mu_a} - \frac{(x_i - \mu_a)^2}{2\mu_a^2} + o\left((x_i - \mu_a)^2\right)\right) \\
&= \log \mu + \frac{1}{\mu}\underbrace{\left(\frac{1}{n}\sum_{i=1}^n x_i - \mu_a\right)}_{\text{equal to } 0} + \frac{1}{2\mu_a^2}\underbrace{\left(\frac{1}{n}\sum_{i=1}^n (x_i - \mu_a)^2\right)}_{\sigma^2} + o\left(\frac{1}{n}\sum_{i=1}^n (x_i - \mu_a)^2\right) \\
&= \log \mu - \frac{\sigma^2}{2\mu^2} + o(\sigma^2).
\end{aligned}
$$

Therefore:

$$\mu_a - \mu_g = \mu_a \left(1 - \exp\left(-\frac{\sigma^2}{2\mu^2}\right)\right) + o(\sigma^2). \tag{8}$$

When $\frac{\sigma^2}{2\mu^2}$ is small, this can be approximated as:

$$\Delta \approx \frac{\sigma^2}{2\mu_a}. \tag{9}$$

$\square$

**Theorem A1** (Scaling Law for Downstream Task Performance). *Consider a language model $M_C$ trained with compute budget $C$ and a set of downstream tasks $\mathcal{P}$. Under the following assumptions: Assumption 1 (Power-law scaling of answer loss): the expected answer loss follows:*

$$L_{\mathcal{P}}(C) := \mathbb{E}_{(q,a_{true})\sim\mathcal{P}}[L(q, a_{true}; C)] = \alpha C^{-\beta} + \gamma, \tag{10}$$

*where $\alpha, \beta, \gamma > 0$ are task-specific constants, with $\gamma$ representing the irreducible loss.*

*Assumption 2 (Unique deterministic answers): Each question has a unique deterministic answer. The model receives score 1 if and only if $M_C$ outputs $a_{true}$, and 0 otherwise.*

*Assumption 3 (Accuracy decomposition): The expected accuracy decomposes as:*

$$\mathbb{E}_{T\sim\mathcal{P}}[\mathrm{Acc}(C)] = g + (1-g) \cdot \mathbb{E}_{(q,a_{true})\sim\mathcal{P}}[p(a_{true}|q, M_C)], \tag{11}$$

*where $g \in [0, 1]$ is the random guessing baseline.*

*Then, the expected accuracy on task set $\mathcal{P}$ can be modeled as:*

$$\mathbb{E}_{\mathcal{P}}[\mathrm{Acc}(C)] = g + (1-g)\left(\exp\left(-\alpha C^{-\beta} - \gamma\right) + \frac{\sigma_L^2(C)}{2\mu_L(C)}\right) + o\left(\sigma_L^2(C)\right), \tag{12}$$

*where $\mu_L(C) = \mathbb{E}_{(q,a_{true})\sim\mathcal{P}}[L(q, a_{true}; C)]$ is the mean loss and $\sigma_L^2(C) = \mathrm{Var}_{(q,a_{true})\sim\mathcal{P}}[L(q, a_{true}; C)]$ is the loss variance across the task set.*

*Proof.* For a task $T = (q, a_{\text{true}}) \in \mathcal{P}$, under assumption 2, $a_{\text{true}}$ is deterministic and unique, thus $p(a_{\text{true}}|q, M_C)$ can be decomposed into token-wise auto-regressive loss.

$$-\log(p(a_{\text{true}}|q, M_C)) = -\log\left(\prod_{i=1}^{k} p(t_i|q, t_{<i}; M_C)\right) \tag{13}$$

$$= -\sum_{i=1}^{k} \log\left(p(t_i|q, t_{<i}; M_C)\right) \tag{14}$$

$$= L(q, a_{\text{true}}; C). \tag{15}$$

Then take the exponential of both sides, and then take the expectation with respect to different tasks in the evaluation set $p = (q, \text{ans}) \in P$. We note that both $p_{\text{ans}}$ and $\text{loss}_{\text{ans}}$ are functions of $C$.

$$\mathbb{E}_p[p(a_{\text{true}}|q, M_C)] = \mathbb{E}_p[\exp(-L(q, a_{\text{true}}; C)] = \frac{1}{n}\sum_{(q,a_{\text{true}})\in P} \exp(-L(q, a_{\text{true}}; C)). \tag{16}$$

We can adopt Lemma 1 to switch from arithmetic mean to geometric mean of loss, and apply the power law assumption 1.

$$\frac{1}{n}\sum_{(q,a_{\text{true}})\in P} \exp(-L(q, a_{\text{true}}; C)) = \exp\underbrace{\left(-\frac{1}{n}\sum_{(q,a_{\text{true}})\in P} L(q, a_{\text{true}}; C)\right)}_{\text{use loss scaling law}} + \frac{\sigma_L^2(C)}{2\mu_L(C)} + o\left(\sigma_L^2(C)\right)$$

$$\tag{17}$$

$$= \exp\left(-\alpha C^{-\beta} - \gamma\right) + \frac{\sigma_L^2(C)}{2\mu_L(C)} + o\left(\sigma_L^2(C)\right), \tag{18}$$

where $n$ in the number of tasks in $\mathcal{P}$, and $\mu, \sigma^2$ follow definitions in the proposition.

Table A1: Prediction errors (EE↓, FE↓) across criteria of extrapolatable subsets.

| Metric / Task Set | Baseline $(a > 1, b > 0.1, 0 \le c < 1)$ | | Ablate a $(a > 0.5)$ | | Ablate b $(b > 0.05)$ | | Ablate c $(0 \le c < 0.5)$ | | w/o threshold | |
|---|---|---|---|---|---|---|---|---|---|---|
| | EE↓ | FE↓ | EE↓ | FE↓ | EE↓ | FE↓ | EE↓ | FE↓ | EE↓ | FE↓ |
| GSM8k | 0.31 | 2.68 | 0.31 | 2.68 | 0.31 | 2.68 | 0.31 | 2.68 | 4.05 | 4.35 |
| MATH | 0.84 | 0.79 | 0.84 | 0.79 | 0.84 | 0.79 | 1.04 | 0.94 | 0.40 | 0.38 |
| BBH | 0.54 | 0.47 | 0.54 | 0.47 | 0.54 | 0.47 | 0.17 | 2.04 | 5.39 | 5.33 |
| TriviaQA | 1.56 | 1.97 | 1.57 | 1.96 | 1.56 | 1.97 | 1.05 | 1.42 | 2.82 | 3.77 |
| MBPP | 1.61 | 2.42 | 1.61 | 2.42 | 1.61 | 2.42 | 1.61 | 2.42 | 1.55 | 1.73 |
| AGIEval | 1.11 | 1.64 | 1.11 | 1.64 | 1.11 | 1.64 | 0.08 | 1.35 | 2.24 | 2.38 |
| DROP | 1.44 | 1.05 | 0.24 | 1.05 | 0.29 | 0.96 | 1.71 | 3.59 | 2.17 | 1.83 |
| MMLU-pro | 1.26 | 1.39 | 1.26 | 1.39 | 1.26 | 1.39 | 2.88 | 3.27 | 1.00 | 1.10 |
| Mean | 1.08 | 1.55 | **0.94** | 1.55 | **0.94** | **1.54** | 1.11 | 2.21 | 2.45 | 2.61 |
| Max | **1.61** | **2.68** | **1.61** | **2.68** | **1.61** | **2.68** | 2.88 | 3.59 | 5.39 | 5.33 |

Finally, we use assumption 3 to align the answer passrate and the accuracy metric.

$$\mathbb{E}_{T \sim \mathcal{P}}[\mathrm{Acc}(C)] = g + (1 - g) \cdot \mathbb{E}_{(q, a_{\text{true}}) \sim \mathcal{P}}[p(a_{\text{true}} | q, M_C)] \tag{19}$$

$$= g + \frac{(1 - g)}{n} \sum_{(q, a_{\text{true}}) \in P} \exp(-L(q, a_{\text{true}}; C)) \tag{20}$$

$$= g + (1 - g) \left( \exp\left(-\alpha C^{-\beta} - \gamma\right) + \frac{\sigma_L^2(C)}{2\mu_L(C)} \right) + o\left(\sigma_L^2(C)\right) \tag{21}$$

$\square$

**Rationality of Assumption 3** Assumption 3 is designed to accommodate tasks with finite answer sets. For such tasks, when calculating $\mathrm{Acc}(C)$, possibilities outside the answer set are disregarded. When $p(a_{\text{true}} | q, M_C)$ approaches 0, $\mathrm{Acc}(C)$ is at the level of a random guess, $g$. When $p(a_{\text{true}} | q, M_C)$ approaches 1, $\mathrm{Acc}(C)$ is close to $p(a_{\text{true}} | q, M_C)$. This assumption implies a linear relationship between $\mathrm{Acc}(C)$ and the probability in the $(0, 1)$ interval. The theorem itself is also effective for tasks with open answer sets, where the probability of a correct random guess can be assumed to be 0 (i.e., $g = 0$).

## C  ADDITIONAL ABLATION STUDIES

### C.1  THE CRITERIA FOR EXTRAPOLATABLE SUBSETS

The criteria for fitting the extrapolation formula (Eq. (4)) are designed to ensure the following:

1. $a > 0$ and $b > 0$: These ensure that accuracy is an increasing function of compute. Larger values of a and b signify that task performance scales more distinctly with compute, leading to fitting curves with better scaling properties and differentiability.

2. $c \ge 0$: This ensures the extrapolated curve's maximum value is less than or equal to 1. An excessively large c implies that the fitting curve has a very low ceiling, which is characteristic of task subsets with poor scaling properties. These are thus considered non-extrapolatable.

We conducted an ablation study on these parameters, as shown in Tab. A1. Starting from our baseline criteria ($a > 1, b > 0.1, 0 \le c < 1$), we individually relaxed the constraints on a, b, and c, and also observed the effect of removing the thresholds entirely.

When the thresholds are removed entirely, the prediction performance degrades significantly. This is because numerous task clusters with poor scaling properties are included in the extrapolation, impairing the overall result. In contrast, individually relaxing the thresholds for a, b, or c still largely preserves the integrity of the filtering criteria. The performance shows remains nearly identical or only slightly decreases compared to the baseline, indicating that while the filtering step is important, our method is not overly sensitive to the specific threshold values.

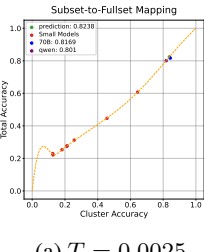 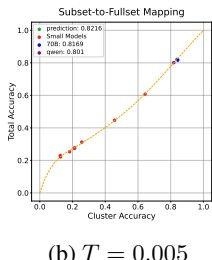 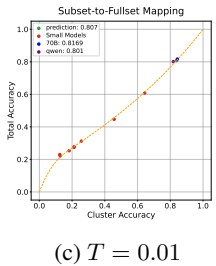 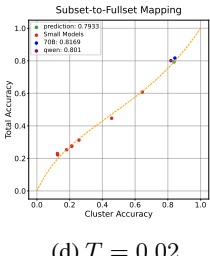

(a) $T = 0.0025$      (b) $T = 0.005$      (c) $T = 0.01$      (d) $T = 0.02$

Figure A1: Ablation of RMSE Threshold $T$.

## C.2   MAPPING METHOD

In the Mapping stage, we map the metrics of the predictable subset to those of the full set, where a cubic smoothing spline is employed to model this relationship. During fitting, we adjust the noise scale $\sigma$ to control the number of segments in the spline. After fitting, we calculate the root mean square error (RMSE). If the RMSE is below a predefined threshold, the fitting process terminates; otherwise, we further decrease $\sigma$ to induce more segments until the RMSE threshold is met.

We demonstrate the impact of different RMSE thresholds on the fitting performance in Fig. A1. It can be observed that the curve exhibits significant overfitting when $T = 0.0025$, whereas the fitted curve deviates from the target points when $T = 0.02$. Consequently, we uniformly adopt $T = 0.005$ as the RMSE threshold across all evaluation sets to achieve the optimal fitting performance.

## C.3   INCORPORATING ANCHOR POINT IN INTERPOLATION MAPPING

We find that the mapping relationship from predictable subset metrics to full evaluation set metrics is similar across models with different training data and architectures. This allows leveraging pre-trained models as "anchors" to refine the mapping and improve final estimation accuracy. In practice, we simply use an open-source model (Qwen2-72B) as a refinement anchor. We first derive an interpolation curve using only small model metrics and fixed points $(0, 0)$, $(1, 1)$, then assess anchor compatibility. This shared mapping implies estimable subset metrics are highly correlated with full-set metrics and less prone to interference from other model parameters than loss-intermediate predictions.

We test two configurations:

- COD (w/o anchor): Full COD pipeline, but no anchor points in the mapping phase;
- COD (w. anchor): COD method with Qwen2-72B as a refinement anchor;

We list the performance of anchor models and the target model in Tab. A2. In Fig. A2, we also compare the differences of using anchor points in the Mapping stage against not using them on two evaluation datasets: MATH and MMLU-pro. The MATH dataset shows a clear discrepancy between the predictable subset and the full set, and adding anchor points significantly improves the fitting and prediction performance. In contrast, the MMLU-pro dataset has a high proportion of predictable instances (listed in Sec. 5.3.3), and its metrics for the predictable subset are close to those of the full set, so introducing anchor points yields little difference.

Table A2: Performance comparison among the target model and anchor models.

| Model | GSM8K | MATH | BBH | TriviaQA | MBPP | AGIEval | DROP | MMLU-pro |
|---|---|---|---|---|---|---|---|---|
| 70B-Dense | 88.55 | 48.02 | 81.69 | 80.66 | 68.00 | 58.20 | 76.82 | 57.28 |
| Qwen2-72B | 88.63 | 53.08 | 80.10 | 84.23 | 71.60 | 64.16 | 77.56 | 56.93 |

To facilitate a direct comparison, we also apply the anchor point to baseline methods including End-to-end(exp) and End-to-end(BNSL), using the same anchor as in the COD method. In practice, the anchor points are directly incorporated into the fitting process of the extrapolation formula.

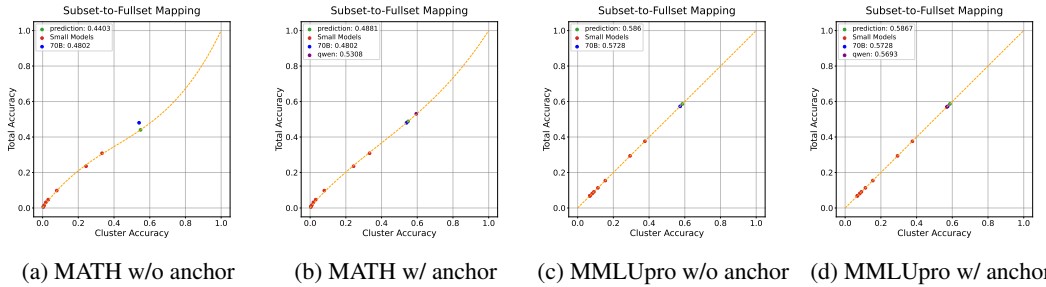

| (a) MATH w/o anchor | (b) MATH w/ anchor | (c) MMLUpro w/o anchor | (d) MMLUpro w/ anchor |

Figure A2: Effectiveness of mapping with or without anchor points.

Table A3: Influence of anchor points in the mapping stage on prediction error (%). Errors < 2% are considered accurate (green), while errors > 5% are considered invalid (red). ↓ indicates lower is better.

| Setting | Method | Overall | | Individual Task Sets | | | | | | | |
|---|---|---|---|---|---|---|---|---|---|---|---|
| | | Mean↓ | Max↓ | GSM8k | MATH | BBH | TriviaQA | MBPP | AGIEval | DROP | MMLU-pro |
| End-to-end(exp) | w/o anchor | 3.10 | 6.00 | 4.00 | 3.86 | 0.64 | 0.68 | 1.75 | 6.00 | 4.11 | 3.72 |
| | w/ anchor | 3.08 | 6.42 | 0.56 | 5.75 | 0.93 | 2.25 | 2.82 | 6.42 | 1.10 | 4.80 |
| End-to-end(BNSL) | w/o anchor | 5.17 | 13.05 | 4.23 | 5.88 | 13.05 | 5.86 | 2.55 | 0.82 | 1.53 | 7.42 |
| | w/ anchor | 3.60 | 6.65 | 0.18 | 5.77 | 1.28 | 4.17 | 4.22 | 6.65 | 1.43 | 5.10 |
| COD | w/o. anchor | 2.65 | 4.98 | 3.10 | 3.99 | 0.97 | 2.38 | 1.59 | 4.98 | 2.86 | 1.32 |
| | w. anchor | **1.55** | **2.68** | 2.68 | 0.79 | 0.47 | 1.97 | 2.42 | 1.64 | 1.05 | 1.39 |

Tab. A3 shows that the COD method incorporating anchor consistently enhances prediction accuracy. However, the End-to-end(exp) and End-to-end(BNSL) baselines failed to derive any benefit from the addition of anchor points. This suggests a stable correlation between the predictable subset and full-set metrics across diverse models, enabling the use of existing model evaluations to improve predictions for new models. In contrast, End-to-end(exp) and End-to-end(BNSL) treat anchor points merely as fitting samples, aligning the prediction target solely with the scaling trend of the anchors. Yet, scaling trends differ significantly across models trained on different data and architectures, manifesting as high variance across different capability dimensions; consequently, these methods fail to produce effective estimations.

Furthermore, since our clustering identifies intrinsic properties of evaluation sets, the derived predictable subsets are applicable to new models.

# D EXPERIMENTAL SETTINGS AND TRAINING RECIPE

**Training recipe.** To establish performance predictions for large language models, we conduct systematic experiments with a suite of smaller-scale models across different parameter counts. All our models are trained from scratch on a corpus of text data. We do not fix the data budget for all models; instead, we maintain a consistent Data-to-CPT (Compute Per Token) ratio for all models, as mentioned in Sec. 5.1. We list model configurations in Tab. A4.

We adopt the in-house training data that comprises multilingual text corpora, with increased weighting for domains such as STEM, code, and general knowledge, following Llama3 (Grattafiori et al., 2024), Deepseek-v2 (Liu et al., 2024), Fineweb-EDU (Lozhkov et al., 2024), etc. We apply several de-duplication methods and data cleaning mechanisms to each data source to ensure high-quality tokens.

The model architecture is consistent with Llama3.1 (Grattafiori et al., 2024), incorporating Grouped-Query Attention (GQA) (Ainslie et al., 2023), SwiGLU activation function (Shazeer, 2020), RM-SNorm (Zhang and Sennrich, 2019) with Pre-normalization, etc. The models are trained using $BF16$ precision with a sequence length of $8192$ and a RoPE Su et al. (2024) base of $500,000$. We employ the AdamW optimizer with $\beta = (0.9, 0.95)$, a weight decay of $0.1$, and a dropout rate of $0.1$.

Table A4: Model architecture specifications across different sizes.

| | 122M | 238M | 411M | 652M | 973M | 1.9B | 7B | 12B | 70B (Target) |
|---|---|---|---|---|---|---|---|---|---|
| Param. (M) | 122 | 238 | 411 | 652 | 973 | 1,901 | 6,980 | 12,022 | 68,452 |
| Compute Per Token (B) | 1.535 | 2.684 | 4.275 | 6.378 | 9.060 | 16.436 | 54.761 | 91.609 | 475.131 |
| Tokens (B) | 26 | 45 | 72 | 108 | 153 | 277 | 923 | 1,544 | 8,012 |
| Continue-Trained Tokens (B) | 3 | 5 | 8 | 12 | 18 | 33 | 114 | 191 | 1,000 |
| Model Dimension | 1,024 | 1,280 | 1,536 | 1,792 | 2,048 | 2,560 | 4,096 | 4,608 | 8,192 |
| FFN Dimension | 3,584 | 4,480 | 5,376 | 6,272 | 7,168 | 8,960 | 14,336 | 16,128 | 28,672 |
| Heads | 8 | 10 | 12 | 14 | 16 | 20 | 32 | 36 | 64 |
| KV Heads | 8 | 10 | 12 | 14 | 16 | 20 | 8 | 12 | 8 |

Table A5: Information of evaluation datasets.

| Dataset | Domain | #Questions | #Shots |
|---|---|---|---|
| GSM8K | Math | 1,319 | 8 |
| MATH | Math | 5,000 | 4 |
| BBH | Reasoning | 6,511 | 3 |
| TriviaQA | Knowledge | 17,944 | 5 |
| MBPP | Coding | 500 | 3 |
| AGIEval | Comprehensive | 8,063 | 5 |
| DROP | Reading | 9,536 | 3 |
| MMLU-pro | Comprehensive | 12,032 | 5 |

All models are trained on a constant learning rate scheduler with a few-step warmup stage. To determine the learning rate and batch size, we adopt the hyperparameter scaling laws from Liu et al. (2024). Specifically, the optimal learning rate $\eta_{\mathrm{opt}}$, and the optimal batch size $B_{\mathrm{opt}}$ are defined as power laws of the compute, measured in FLOPs: $\eta_{opt} = a_1 \cdot C^{-b_1}$ and $B_{opt} = a_2 \cdot C^{b_2}$, where $a_1$, $b_1$, $a_2$, $b_2$ are parameters to be fitted. We perform a grid search on our small models to identify their optimal learning rates and batch sizes, and then extrapolate these findings to the bigger models.

**Training Resources.** The 7B dense model is trained on 923B tokens, consuming 52,800 H800 GPU-hours. The computational resources used for the other models can be estimated proportionally based on their respective compute requirements.

**Evaluation settings and protocol.** We conducted performance scaling estimation experiments across eight major LLM evaluation sets. These evaluation sets span a diverse range of capabilities, including Math, Reasoning, Knowledge, Coding, Reading, and general abilities. All pretrained LLMs were evaluated using a few-shot methodology to obtain the performance metrics. Detailed information is provided in Tab. A5. Our evaluation methodology aligns with that used for the Llama3 (Grattafiori et al., 2024) pre-trained models. We assess the models' capabilities directly through few-shot text completion tasks without any instruction tuning or Supervised Fine-Tuning (SFT). This evaluation method is chosen because even a small amount of SFT data can significantly influence performance on downstream tasks, thereby not reflecting the inherent capabilities of the pre-trained model itself.

**Software Framework.** All models are trained using the Megatron framework. The evaluation code is an in-house implementation designed to be consistent with the Llama3 (Grattafiori et al., 2024) evaluation methodology.

# E PERFORMANCE PREDICTION FOR CONTINUE-PRETRAINED LLMS

Leading industry pre-trained LLMs (e.g., Deepseek-v3 (DeepSeek-AI et al., 2025), Llama3 (Grattafiori et al., 2024), Qwen-2.5 (Yang et al., 2024a)) adopt the Continual Training (CT) strategy of concentrating high-quality data towards the end of the pre-training process. This phase is typically accompanied by learning rate decay, enabling the model to fully absorb this high-quality data. Due to significant changes in data distribution and the learning rate schedule, this approach often yields substantial improvements in metrics. Predicting a large model's final capability based solely on its performance during a "stable" phase with consistent data distribution does not

reflect its ultimate capability. Therefore, we supplement this by providing metric predictions for the high-quality CT phase.

The relationship between model parameter scale and the volume of CT tokens is listed in Tab. A4. We conduct the same COD pipeline for CT models. We control the data distribution of the stable and decay phases for various smaller models, as well as their token-to-parameter ratio, to be consistent with the large model targeted for prediction. The last checkpoint is used for evaluation. Based on prior clustering labels, we perform fitting, extrapolation, and mapping to obtain the predicted performance for the large model.

Table A6: Predicted vs. actual metrics for an LLM with 70B parameters after high-quality continued pretraining. Errors < 2% are considered accurate (green)

| Evaluation Set | Predicted Metric | Actual Metric | Prediction Error |
|---|---|---|---|
| GSM8k | 93.10 | 91.81 | 1.29 |
| MATH | 56.35 | 52.68 | 3.67 |
| BBH | 83.05 | 85.32 | 2.27 |
| TriviaQA | 79.29 | 84.05 | 4.76 |
| MBPP | 72.42 | 73.20 | 0.78 |
| AGIEval | 63.22 | 64.18 | 0.96 |
| DROP | 82.34 | 81.39 | 0.95 |
| MMLU-pro | 62.11 | 59.34 | 2.77 |

Results listed in Tab. A6 show that the proposed COD method achieves an average prediction error of 2.18%. We observe that MATH and TriviaQA exhibit relatively large prediction errors. We hypothesize that there are two main categories of reasons for this inaccuracy:

1. The CT data and the evaluation sets possess a significant correlation. For example, in math-related evaluation sets, a modest amount of training can yield substantial improvements in performance metrics. In such scenarios, the metrics for smaller models tend to show greater volatility and have inaccurate evaluations.

2. The CT data exhibits inherent distribution bias, such that certain evaluation sets, such as TriviaQA, do not derive performance gains from it. This leads to potential significant fluctuations in the metrics after the CT phase, thereby diminishing the accuracy of extrapolating to larger models.

## F    DIFFICULTY DISTRIBUTION OF PREDICTABLE SUBSET

We analyze the proportion of predictable subset tasks across different difficulty levels. The difficulty distributions of predictable subset versus complete sets for different evaluation benchmarks are illustrated in Fig. A3. We use the scores from the 12B model as the basis for difficulty classification. The results show that MMLU-pro and GSM8k evaluation sets have larger proportions of predictable subsets, indicating that most questions in these datasets exhibit good performance scaling properties. In contrast, many difficult questions with near-zero scores in the MATH evaluation set fall outside the predictable subset, requiring adjustment during the mapping phase. Meanwhile, BBH exhibits consistent proportions of its predictable subset across varying difficulty levels, as some questions display oscillatory patterns with limited improvement, even with increased computational resources.

The proportion of the predictable subset can serve as a metric for assessing evaluation set quality. Evaluation sets with larger predictable subsets yield more reliable experimental conclusions from smaller models. When constructing evaluation sets, we recommend screening or supplementing unpredictable clusters and ensuring a minimum number of questions for each difficulty feature to reduce metric volatility.

## G    COMPUTATIONAL COST

The extra computational overhead of running COD is not expensive compared to training a series of small models with increasing parameter sizes. The main additional cost comes from performing 100

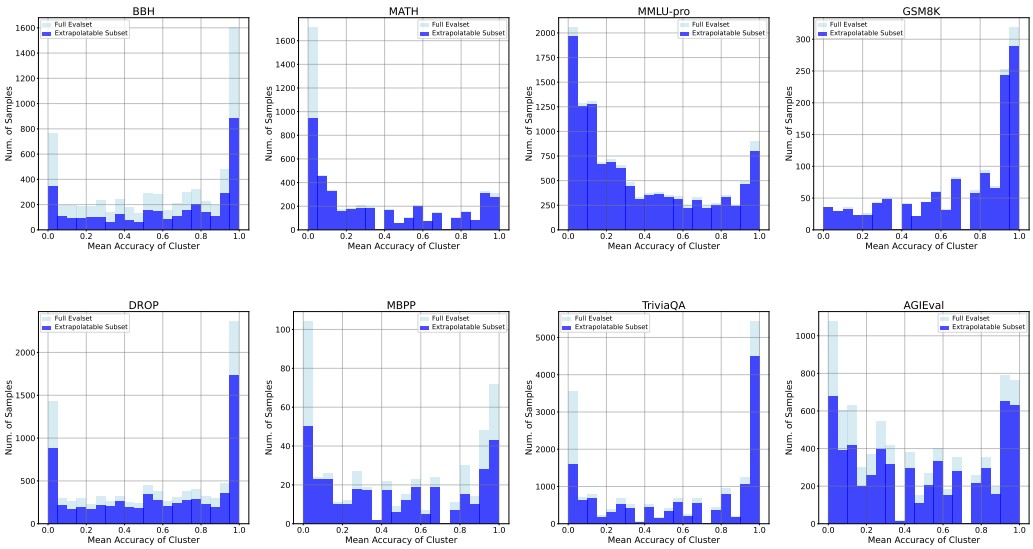

Figure A3: Difficulty distribution comparison on a 12B model between predictable subset and full evaluation set.

inference evaluations on the evaluation set for each small model (the cost of clustering algorithm is negligible compared to the inference cost). The computational complexity is $O(TMN)$, where $T$ is the number of evaluation runs, $M$ is the number of tokens for one evaluation, and $N$ is the maximum parameter size of the small models used for prediction. The corresponding token usage is $O(TM)$.

In particular, for an evaluation set requiring $1M$ tokens, a total of $100M$ tokens for small model inference is needed. In our experiments, the training token count for the 12B small model is $1.554T$. Considering that a training token is typically 3 times more costly than an inference token, the additional cost of COD is approximately $\frac{100M}{1.554T*3} \approx 0.002\%$ of the training cost.

## H    LIMITATIONS

**Compromised robustness due to excessive hyperparameter.** The complete pipeline of our proposed COD method incorporates several hyperparameters designed to constrain and refine the outcomes of various stages. These include the minimum intra-cluster sample size $K$, the adaptive bandwidth hyperparameter $Q$, and the maximum intra-cluster distance threshold $U$ for the clustering phase; parameters $a, b$, and $c$ for filtering extrapolatable subsets during curve fitting; and the RMSE threshold $T$ utilized during the mapping process.

Regarding the clustering-related hyperparameters, they can be omitted if pre-computed cluster assignments are reused; otherwise, we provide empirically validated default values as reliable priors. For the remaining hyperparameters, we present comprehensive ablation studies in the paper, demonstrating that the final predictive performance is robust and relatively insensitive to these settings. Despite these mitigations, the reliance on a multi-parameter configuration may pose challenges to the COD method's ease of deployment when generalizing to novel prediction scenarios.

**Category of evaluation sets.** The proposed Clustering-on-Difficulty method requires a sufficient number of test cases, as too few samples can lead to unstable cluster metrics and ineffective estimation. From an evaluation set design perspective, an evaluation set with good predictive properties enables more effective generalization from small-scale to large-scale models, thus providing better guidance for model iteration.

Furthermore, we have not included multiple-choice tasks that require comparing the logits of correct options to calculate scores. These tasks creating a discrepancy between the answer loss and the model's true passrate, which violates the assumptions of the proposed Scaling Law for downstream task performance.

**The prediction accuracy for smaller models is unsatisfactory.** Since our proposed COD (Clustering on Difficulty) method involves modeling the scaling of sample difficulty within the evaluation set, the clustering process requires a certain scale of models to participate in pass rate evaluation. This ensures an accurate estimation of sample difficulty. However, when predicting the performance of relatively smaller target models (e.g., around 10B parameters), the proxy models used for clustering and fitting are typically limited to even smaller scales (e.g., 2B). In such scenarios, a significant portion of samples may exhibit non-emergent or nascent emergent behaviors, making it challenging to accurately model difficulty features that scale with compute. Under these specific constraints, methods that perform sample-wise extrapolation—such as PassUntil (Hu et al., 2024)—tend to yield more robust predictive performance. Nonetheless, the primary utility of metric prediction lies in forecasting the downstream performance of significantly larger models. From this perspective, our COD method maintains broad applicability and significant value in mainstream scaling law research.

**Chain-of-thought performance prediction**. Theorem 1 assumes that evaluation sets directly assess models' ability to provide answers. However, increasingly more evaluations allow models to think before providing answers. Recent works on inference time scaling (Snell et al., 2024; Bansal et al., 2024) further demonstrate that for tasks involving mathematics, reasoning, and coding, training models to complete tasks through longer inference computation can significantly improve downstream task performance. In cases where the reasoning process or answers are not unique, the relationship between a model's answer loss and passrate on a task may not necessarily follow the exponential relationship between the answer loss and the sample passrate. Although our COD framework still achieves reasonable prediction performance in such scenarios, its theoretical foundation lacks sufficient explanation for the performance scaling of chain-of-thought based tasks. Therefore, we consider improving prediction methods based on chain-of-thought characteristics and expanding theoretical foundations as future work.

