# OpenReview forum: "Unveiling Downstream Performance Scaling of LLMs: A Clustering-Based Perspective"
_ICLR.cc/2026/Conference — ICLR 2026 Poster_

### Official Review · Reviewer_DRNX · 2025-10-30

**Soundness:** 2
**Presentation:** 3
**Contribution:** 3
**Rating:** 4
**Confidence:** 4

**Summary:**

This paper introduces a cluster-based method for task scaling law (i.e., predicting larger model performance based on smaller model performance). The method has four steps: (1) cluster test examples based on their difficulty features (i.e., how small language models perform on the example); (2) fitting scaling law for each cluster; (3) extrapolation for each cluster; (4) mapping back to the full evaluation set performance. On an internal model series, the proposed method outperforms baselines such as end-to-end prediction or loss-intermediate prediction. Further ablation is done on clustering methods and scaling law formula.

**Strengths:**

* The paper presents an effective and reasonable approach for downstream task scaling law prediction. The pilot study points out limitations about several questionable assumptions in task scaling law, which I found insightful. The proposed method is reasonable and intuitive.

**Weaknesses:**

* Some details of the proposed method is unclear or missing. See questions below.
* Results were limited to one model series. The paper could be more convincing if the method was shown to be effective on various model families.

**Questions:**

* Line 151: "three random clusters" - does this mean data points are randomly assigned to clusters, or three clusters randomly selected from your work in later chapters? Very confusing at this point.
* Line 214: "we generate 100 samples" - does this mean sampling 100 answers for every input query?
* Line 337: "we calibrate ... using evaluation results from existing models as anchors". Does this mean you fit the parameters in Eq. (5) using another model's performance? After reading appendix C.3, it seems this anchoring step is critical as removing it will results in deteriorated performance (worse than the end-to-end baseline). I have three follow-up questions regarding this: (1) Do baseline methods have access to this extra information? (2) Can you elaborate more on the in-house MoE model you used here? How large is it? (3) Can you report the performance of the calibration model and the target model (70B) on the 8 tasks in Table 1?

(I'm willing to adjust my rating if the questions above are resolved.)

Suggestions:
* I was very confused when reading the abstract. Terminologies like "difficulty scaling features" and "support subset" were not defined. "hindering prediction by smaller models" is confusing, do you mean "extrapolation from performance using smaller models"? "the exclusion of tasks" - do you exclude tasks or test examples? (My current understanding is that you exclude some test examples.)
* Suggested reference:
  * Establishing Task Scaling Laws via Compute-Efficient Model Ladders (https://arxiv.org/abs/2412.04403)
  * Observational Scaling Laws and the Predictability of Language Model Performance (https://arxiv.org/abs/2405.10938)
  * How Predictable Are Large Language Model Capabilities? A Case Study on BIG-bench (https://arxiv.org/abs/2305.14947)

---

> ### Author Response · Authors · 2025-11-23
>
> We strictly appreciate your detailed review. We have organized our responses below.
>
> ### **For Question 1**
>
> The phrase "three random clusters" refers to "three clusters randomly selected from the analysis presented in later chapters." We apologize for the confusion and clarify this definition explicitly in line 161.
>
> ### **For Question 2**
>
> Yes. For every evaluation query, we generate 100 samples to calculate its **Pass Rate (Pass@100)**. This is done to accurately characterize the difficulty profile and granular information of each specific query.
>
>
> ### **For Question 3**
>
> We did not include anchor points in the baseline methods for the following reasons:
>
> 1.  **Loss-intermediate Method:** This method relies on training loss or in-distribution (ID) validation loss as an intermediate metric. For models trained on data with different distributions (OOD), the loss values are not quantitatively comparable. Therefore, open-source OOD models cannot serve as anchors. Furthermore, the ID MoE model used in our work underwent **continued pre-training**, meaning its training data is not strictly aligned with the small models, rendering it unsuitable as an anchor for loss extrapolation.
> 2.  **End-to-end Method:** This method directly models the quantitative relationship between **Performance** and **Compute**. Consequently, the data points used for fitting must maintain consistency in how model capability is accumulated; otherwise, extrapolation bias occurs. For example, training with higher-quality data or using MoE architectures can achieve the same or higher performance with significantly less compute. This shifts the anchor point away from the original scaling law curve.
>
> To empirically validate this, we conducted experiments adding the same anchor points used in COD to the fitting stages of the **End-to-end** and **BNSL** methods. We add this experiment in Appendix C.3 (Table 3). The results demonstrate that adding extra anchors does not effectively improve prediction performance of the End-to-end method. The BSNL method benefits from the introduction of anchor points but still underperforms end-to-end approaches.
>
> * **Impact of ID Anchor (MoE):** Due to the higher computational efficiency of MoE, it achieves high performance with less compute. Incorporating this into a Compute-Performance fit misleads the extrapolation, resulting in severe prediction errors.
> * **Impact of OOD Anchor:** While OOD anchors occasionally improve precision on specific benchmarks (because their compute/performance happens to align with the target), the difference in training data distribution means the model capabilities are not consistent across all dimensions. Once the anchor's performance deviates from the target model's trajectory on a specific task, the fitted curve becomes biased, harming prediction accuracy.
>
> Fundamentally, The **COD method** succeeds where baselines fail because the **mapping process** models the relationship between the *subset scores* and the *full-set scores*. This relationship is less sensitive to training data efficiency or model architecture, allowing the anchor point to effectively guide the curve fitting.
>
> **Model Parameters for the MoE Anchor:**
>
> | Model Dimension | Layers | Heads | KV Heads | Expert Dim | Total Experts | Activated Experts | Tokens | Activated Param | Total Param |
> | :---: | :---: | :---: | :---: | :---: | :---: | :---: | :---: | :---: | :---: |
> | 2048 | 40 | 16 | 4 | 1024 | 128 | 8 | 23T | 2.7B | 33B |
>
> **Comparison of Anchor vs. Target Performance:**
> We also provide a comparison of the Anchor Models and the Target Model across 8 benchmarks (Appendix C.3)
>
> The results show that our Anchor Points differ significantly from the target 70B model on several benchmarks, and the direction of capability variance is inconsistent (e.g., Qwen2-72B scores 5.96 points *higher* on AGIEval but the 2.5B-MoE scores 8.49 points *lower* on TriviaQA). Thus, simply substituting the target with an anchor does not yield a valid prediction.
>
>
> ### **For Suggestions**
>
> * **"Difficulty scaling features":** This refers to the vector constructed by concatenating the Pass Rates of a series of small models (Method section).
> * **"Support subset":** This refers to the **extrapolatable subset** obtained after our filtering process.
> * **"Hindering prediction by smaller models":** This specifically refers to "hindering **extrapolation** from performance using smaller models."
> * **"The exclusion of tasks":** This refers to the exclusion of **non-emergent test examples** when predicting the extrapolatable subset. However, our final prediction for the full evaluation set is obtained by establishing a mapping between the metrics of the extrapolatable subset and the full set.
>
> ### **Adding Citations**
>
> We agree that the referenced literature is highly relevant to our work. We have added citations to these papers in the Related Work section of the revised manuscript.

---

### Official Review · Reviewer_fKzE · 2025-11-01

**Soundness:** 2
**Presentation:** 3
**Contribution:** 1
**Rating:** 2
**Confidence:** 4

**Summary:**

The paper proposes Clustering-On-Difficulty (COD) to predict the downstream accuracy of a large model from smaller models. The pipeline is: (i) build a task‑wise pass‑rate vector across model scales and cluster tasks (improved MeanShift with radius & min‑size constraints) to reduce intra‑cluster variance; (ii) fit a cluster‑wise performance–compute scaling law derived from a loss scaling assumption; (iii) filter “non‑extrapolatable” clusters via parameter thresholds; (iv) map the predicted accuracy on the “predictable subset” to the full evaluation set with a monotone polynomial (quartic by default), optionally calibrated with anchor models. On eight eval sets (GSM8K, MATH, BBH, TriviaQA, MBPP, AGIEval, DROP, MMLU‑pro), COD reports 1.36% average prediction error across eight key LLM benchmarks, outperforming loss‑intermediate, end‑to‑end, pass‑rate, and BNSL baselines.

**Strengths:**

1. Directly predicting downstream performance (not just loss) is operationally valuable.
2. The author spends much effort on training nine models (122M→70B) under a consistent recipe and eight diverse benchmarks. Such results are valuable for the community.
3. The paper is well-written and easy to understand.

**Weaknesses:**

1. The generalizability of the model is questionable. All selected benchmarks are widely adopted in the community. How does the proposed generalize to unseen benchmarks?
2. The author only compares with their own baselines. Many performance prediction solutions on downstream benchmarks are missing in the comparison.
3. The COD experiments heavily rely on heuristic hyperparameters such as a, b & c. This limits its generalizability to other cases.
4. Typos: in the abstract, the mean error is 1.36%, in Table 1, it is 1.63%. Which one is correct?

**Questions:**

See cons

---

> ### Author Response · Authors · 2025-11-23
>
> We sincerely appreciate your insightful review comments, which have been instrumental in improving our work. We have organized our responses to the specific **Weaknesses** raised below.
>
> ---
>
> ### **For Weakness 1**
>
> We have presented conclusions on 8 mainstream benchmarks covering various evaluation dimensions, fully describing the model's capabilities across different facets. This demonstrates the generalizability of the COD method. We selected these benchmarks because they are the most frequently cited in technical reports and are of the greatest interest to the community, **not** because they share identical properties or offer superior predictability.
>
> In practice, we find that these benchmarks effectively monitor the model pre-training process, serving to guide training decisions and estimate the potential of data and models.
>
> Fundamentally, **unseen benchmarks face the same challenges as mainstream ones**, such as "heterogeneous difficulty scaling laws" and the "lack of emergence in small model capabilities." Therefore, our method is equally applicable to other benchmarks. Due to time constraints, we could not generate results for additional benchmarks within the rebuttal period. However, we plan to open-source the complete code framework in the future and welcome researchers to evaluate our method on a wider range of benchmarks.
>
> ---
>
> ### **For Weakness 2**
>
> The baselines we adopted represent the current mainstream methods. Research in this area is relatively limited. To the best of our knowledge, existing mainstream methods can be divided into two categories: The first category proposes an end-to-end downstream task prediction formula and provides the interpretability of the formula. The focus of such works lies in finding an ideal formula for modeling the performance scaling trend. The second category leverages the prediction results of the Loss-Scaling Law as references and transforms them into the task of downstream task metric prediction. The formula adopted in this paper is derived from highly recognized works [1][2], ensuring its rationality.
> In addition, since the core benefit of the COD method stems from modeling the scaling law of heterogeneous evaluation metrics, we also introduce the BSNL method based on piecewise function fitting as a reference.
>
> [1] Xiao, Chaojun, et al. "Densing law of llms." Nature Machine Intelligence (2025): 1-11.
>
> [2] Achiam, Josh, et al. "Gpt-4 technical report." arXiv preprint arXiv:2303.08774 (2023).
>
> ---
>
> ### **For Weakness 3**
>
> We acknowledge that the COD method involves multiple stages and requires preset hyperparameters. However, the core concept of COD utilizes **difficulty clustering** to perform component-wise extrapolation on subsets of questions with similar scaling features, followed by a weighted summation.
>
> In the main experiment (Table 1), we present the prediction metrics for **COD (w/o mapping)**—which relies solely on the clustering, extrapolation, and weighted summation pipeline. This metric already achieves prediction accuracy surpassing all baselines, demonstrating that the core difficulty clustering method is effective. At this stage, neither the threshold design for $a, b, c$ nor the fitting in the mapping phase is required.
>
> To further demonstrate the method's robustness and generalizability, we conducted a detailed Ablation Studies:
>
> 1.  In Appendix C.1 and Table A1, we ablated the parameter ranges for $a, b, c$ used to filter the extrapolatable subset. The results indicate that adjustments to these three parameters have minimal impact on the final prediction. The core objective here is to find a quantitative method to filter out non-extrapolatable subsets. These subsets have relatively obvious scaling features (performance near 0 or oscillating), making the filtering process insensitive to specific threshold values.
> 2.  We applied the proposed method to predict the performance of **Continually Pretrained LLMs** (updated Appendix E) and other model sizes (using models $<1.9$B to predict $12$B, updated Section 5.3.1). In all configurations, the COD method achieved optimal results, demonstrating its robustness.
>
> In summary, the core of our work is characterizing the **heterogeneous metric scaling structure** within the evaluation set via difficulty clustering. This is not strictly dependent on the specific model series being predicted and thus possesses generalizability. In practice, when facing a newly trained model series, we can utilize pre-calculated cluster labels for prediction, incurring no additional inference cost compared to baseline methods. We will open-source the code for cluster centers and extrapolation calculations to improve usability.
>
> ---
>
> ### **For Weakness 4**
>
> This is a typo. We have corrected it in the updated manuscript.
>
> **If the reviewer agrees with our clarification above and kindly re-evaluates the value of our work, we would be very grateful. We are happy to address any further questions regarding our work.**

---

### Official Review · Reviewer_D8BZ · 2025-11-03

**Soundness:** 4
**Presentation:** 2
**Contribution:** 3
**Rating:** 8
**Confidence:** 5

**Summary:**

## Summary

This work  proposes a new framework for predicting downstream performance using smalller proxy models. It introduces Clustering-On-Difficulty (COD), which groups samples in eval tasks by estimated difficulty  to reduce variance and enable more stable predictions￼. The paper provides theoretical analysis to justify this approach, and shows, thorugh extensive experimental results, that their approach outperforms other competing methods. Overall it is a strong work, and should be, in my opinion, accepted to ICLR 2026.

## Typos, missing references
- line 61: " 1.36%ha"
- [Bhagia et al 2024](https://arxiv.org/abs/2412.04403)

**Strengths:**

- Method is effective, well justified, and works across large set of taks.
- The emprical results are very convincing.
- The paper includes extensive ablations in main text and appendix.
- Paper is important contribuition in the field of estimating downstream performance, as it significantly improve ability to predict downstream performance.

**Weaknesses:**

- The core for the clustering approach seems to hinge on the estimation of difficulty per each instance in a benchmark. To do so, the paper seems to imply that models that are also target of scaling laws are used. Unclear if the proposed approach would work if using a different model family (i.e., how would this approach work if one has to bootstrap from external open-weights LLMs?)
- To estimate task difficulty, it seems that one need a model already trained at largest target size to build clustering features? Is that the case?
- The paper is missing an ablation showing how prediction accuracy changes if fewer/smaller models are used.
- The main text is really difficult to follow due to missing details. The details are generally provided in the appendix, but the back-and-forth makes following the approach very hard.
- The work is applied to MC tasks only. MC tasks have interesting discuitnuity properties [\(Wiegreffe et al, 2025\)](https://arxiv.org/abs/2407.15018), but are far from the only formulation of interest.

**Questions:**

I've made some educated guesses in my weaknesses section. Would be great if the authors could comment on them to ensure that I'm correct.

---

> ### Author Response · Authors · 2025-11-23
>
> We sincerely appreciate your recognition of the novelty and practicality of our work, as well as your constructive suggestions. The weaknesses and questions you raised are highly valuable to us. We have organized our responses below to address these points.
>
> ---
>
> ### **For Weakness 1**
>
> The heterogeneous scaling trends described by the clustering are intrinsic properties of the dataset itself. These scaling properties are generally transferable across different models. Prediction performance improves when the same model series is used for both clustering and extrapolation. This is because the consistency between the clustering phase and the extrapolation phase is higher, leading to reduced intra-class variance, which aligns more closely with our theoretical assumptions.
>
> ---
>
> ### **For Weakness 2**
> In the **Method** section, we define the prediction task as utilizing evaluation metrics from a series of small models to predict the metrics of large models. In practice, the models used to construct clustering features are consistent with those used for the fitting process; we strictly avoid using larger models to construct features to prevent feature leakage.
>
> Specifically, when predicting metrics for the **70B** model, we exclusively used metrics from models of size **12B and below**. As mentioned in the original Appendix E (Line 1108) and the updated Section 5.3.1 (Line 457), when predicting the **12B** model, we used models under **1.9B**, and the clustering features were constructed using these sub-1.9B models. We have clarified this experimental setting in Section 4.1 (Line 237) of the revised manuscript.
>
> ---
>
> ### **For Weakness 3**
> We originally presented experiments using models of **1.9B and smaller** to predict the **12B** model in **Appendix E**, where the COD method achieved similarly superior performance as in the 70B prediction task. Recognizing the importance of this analysis to the validity of our method, we have relocated this content to Section 5.3.1 of the main text to make it more prominent.
>
> ---
>
> ### **For Weakness 4**
> Thank you for this feedback. We have made significant efforts to optimize the writing structure. We have moved relevant content from the Appendix to the main text and ensured that essential details are clearly articulated within the main body of the paper to improve readability and flow.
>
> ---
>
> ### **For Weakness 5**
>
> In fact, our work is applicable to both Multiple Choice (MC) tasks and Free-form Answer Parsing tasks. The ambiguity likely arose from our statement in the Limitation section: *"we have not included multiple-choice tasks that only require option selection."*
>
> By this, we referred to multiple-choice evaluation methods that do not verify the consistency between the generated output and the answer string, but rather compare the **Logits** of the output options (selecting the option with the maximum Logit as the correct answer). This type of evaluation creates a discrepancy between the "pass rate" and actual accuracy: a score is awarded simply if the correct option's Logit is higher than the others, regardless of the generated text. Therefore, it is incompatible with our prediction method, which is established based on **sampling-based pass rates**.
>
> We have revised the **Limitation** section to state: *"we have not included multiple-choice tasks that require comparing the logits of correct options to calculate scores."*
>
> ---
>
> ### **Typos**
> The value of **1.36%** appearing in the abstract was a typo. The value presented in Table 1 of the main text is accurate. We have corrected this error in the updated Abstract and Introduction.
>
> **We hope you are satisfied with our clarifications above. If you have further comments, and we'd be glad to answer them in the discussion period.**

---

### Official Review · Reviewer_vwUL · 2025-11-03

**Soundness:** 2
**Presentation:** 2
**Contribution:** 3
**Rating:** 6
**Confidence:** 4

**Summary:**

This paper tries to address the challenge of accurately predicting downstream performance of large language models (LLMs) before full-scale training. Existing scaling laws—either based on loss or direct performance–compute curves—fail to capture emergent behaviors and heterogeneous task difficulty.
The authors propose a Clustering-On-Difficulty (COD) framework that groups evaluation samples by their performance-scaling patterns using an improved MeanShift algorithm. Within each cluster, they fit a theoretically derived performance scaling law to extrapolate results from small models to large ones. Predictions from extrapolatable clusters are then mapped via a polynomial function to the full benchmark.
Experiments on eight benchmarks (e.g., MATH, GSM8K, BBH, MMLU-pro) show that COD achieves an average better prediction error of a 70B model than baselines.

**Strengths:**

1. The COD approach provides a clear decomposition of the performance-prediction problem into clustering, cluster-wise fitting, subset extrapolation, and subset-to-full mapping. This layered design could handle emergent and heterogeneous behaviors better.
2. The improved MeanShift with radius and minimum-cluster constraints balances automatic cluster discovery and intra-cluster homogeneity, outperforming K-Means, DBSCAN, and baseline MeanShift on both intra-cluster metrics and prediction error.
3. Eight diverse benchmarks and multiple baselines are covered, including detailed ablations (clustering, extrapolation formulas, mapping functions).
4. The paper provides pseudocode, theoretical proofs, and extensive appendices on hyperparameters, smoothing, and computational cost.

**Weaknesses:**

1. COD introduces multiple stages (feature extraction, clustering, extrapolation, mapping) with several hyperparameters (radius R, min cluster K, thresholds a/b/c, polynomial degree). This may hinder practical adoption and stability.
2. In what extent of error rate, could the preformance prediction be useful? Can this actually reflect the actual final benchmark performance?
3. In introduct the error rate is 1.36%, but in the table of experiment results, the error rate seems to be 1.63, which is inconsistent.

**Questions:**

Some typos: e.g. 1.36%ha in line 061, can them be fixed?

Any specific evidence why this method could handle better emergent and heterogeneous behaviors?

---

> ### Author Response · Authors · 2025-11-23
>
> We sincerely appreciate the insightful reviews, as well as your recognition of our work. We have organized our responses as follows:
>
> ---
> ### **For Weakness 3 & Question 1**
> It was a typo, the actual value is 1.63. We have corrected this in Line 24 of the revised manuscript.
> ### **For Question 2**
> We address these behaviors by dividing the evaluation set into two parts: the **non-emergent subset** and the **extrapolatable subset**.
>
> * **Handling Non-emergent Behaviors:** On the *non-emergent subset*, performance metrics for small models often fluctuate around random guessing or exhibit severe volatility, causing existing single-stage fitting methods to fail. However, our method identifies a strong correlation between the *extrapolatable subset* metrics and the full set metrics. This allows us to effectively estimate the full set performance using the extrapolatable subset. As shown in Figure A1 (Appendix), the relationship between the extrapolatable subset and the full set can be fitted with a smooth curve.
> * **Handling Heterogeneous Behaviors:** On the *extrapolatable subset*, we observe that different problem clusters exhibit distinct scaling laws. In the right panel of Figure 1 (using BBH as an example), we present Accuracy-Compute curves that verify three distinct scaling trajectories. The proposed method extrapolates within clusters first and then superimposes the results, thereby accurately characterizing the heterogeneous scaling behavior internal to the evaluation set.
> (Clarified in Intro, line 59).
>
> ### **For Weakness 1**
>
> To address the concern regarding complexity and stability:
> 1.  **Effectiveness of Core Mechanism:** In the main experiment (Table 1), we present the prediction metrics for **COD (w/o mapping)**—which relies solely on the clustering, extrapolation, and weighted summation pipeline. This simplified version already achieves prediction accuracy surpassing all baselines, demonstrating that the core method of difficulty clustering is effective even without the mapping stage or the $a, b, c$ threshold design.
> 2.  **Robustness (Ablation Studies):** To demonstrate robustness, we ablated the parameter range for $a, b, c$ (used to filter the extrapolatable subset) in Appendix C.1 and Table A1. The results indicate that adjustments to these parameters have minimal impact on the final prediction. The core objective here is to quantifiably filter out non-extrapolatable subsets. These subsets have obvious scaling features (performance near 0 or oscillating), making the method insensitive to specific threshold values.
> 3.  **Practical Usability:** The difficulty clustering is a one-time cost. This clustering characterizes the internal scaling structure of the benchmark and is not strictly dependent on the specific model series being predicted. In practice, when facing a newly trained model series, we can utilize pre-calculated cluster labels, incurring no additional inference cost compared to baseline methods. We will **open-source** the cluster centers and extrapolation code to improve the ease of use.
>
> ### **For Weakness 2**
> We define the tolerance range for prediction error based on the fitting error with small models. Specifically, we calculate the **Prediction Interval (PI)** using the formula:
> $$\text{Range} = \text{Median Absolute Deviation (MAD)} \times t_{\text{score}} (4.604) \times \lambda (2)$$
> , where $4.604$ is the t-score for a 99\% confidence level, and $2$ is the Laplace correction coefficient.
>
> We consider the performance prediction to be accurate and reflective of the model's actual final benchmark performance if the predicted metric falls within this confidence interval.
>
> | Dataset | Prediction Error Tolerance Range | Ground Truth | Prediction Interval (PI) | Loss-intermediate | End-to-end | BNSL | COD (w/o mapping) | COD |
> | :--- | :---: | :---: | :---: | :---: | :---: | :---: | :---: | :---: |
> | **gsm8k** | 5.53 | 88.55 | [83.02, 94.08] | 79.23 | 92.55* | 84.32* | 86.12* | **86.32*** |
> | **MATH** | 1.30 | 48.02 | [46.72, 49.32] | 41.22 | 51.88 | 53.90 | 48.16* | **49.30*** |
> | **BBH** | 8.31 | 81.69 | [73.38, 90.00] | 84.44* | 81.05* | 94.74 | 74.74* | **83.46*** |
> | **TriviaQA** | 4.51 | 80.66 | [76.15, 85.17] | 75.80 | 81.34* | 86.52 | 73.30 | **79.02*** |
> | **MBPP** | 4.10 | 68.00 | [63.90, 72.10] | 62.74 | 66.25* | 70.55* | 67.76* | **70.19*** |
> | **AGIEval** | 2.76 | 58.20 | [55.44, 60.96] | 57.02* | 64.20 | 59.02* | 55.54* | **55.82*** |
> | **DROP** | 4.77 | 76.82 | [72.05, 81.59] | 71.81 | 72.70* | 78.34* | 71.68 | **77.05*** |
> | **MMLU-pro** | 2.12 | 57.28 | [55.16, 59.40] | 52.01 | 61.00 | 64.70 | 58.48* | **58.63*** |
>
> We can see the proposed COD method maintains accurate prediction results (falling within the PI) across all evaluation sets. In contrast, all Baseline methods deviate from the PI range on at least three benchmarks.
>
> **We hope you are satisfied with our response. If you have further comments, and we'd be glad to answer them in the discussion period.**

---

### Meta-Review · Area_Chair_ZEpv · 2026-01-13

**Summary:**

The paper proposes a framework to predict how LLMs will behave in downstream tasks from small proxy models. The method they call Clustering on Difficulty (COD) groups tasks by estimated difficulty and obtains more accurate scaling downstream predictions.

The method is very reasonable and (as far as I know, and also according to the reviewers) novel.
Also the empirical evaluation is very well done and convincing.

One recommendation for the authors:
I believe the technique the authors are inventing is the well-known classical statistical technique of stratification.
Stratification in statistics is the idea of splitting a population into meaningful subgroups (“strata”) e.g. age bands, or income level, sampling these groups and then adjusting the representation of each group in a general population.

Stratification Increases precision and reduces variance. Sampling within each stratum makes estimates more stable, especially when strata are internally similar.

 I think this paper is basically a stratification by difficulty and possibly classical statistical techniques can be deployed. At the very least, I would propose that the authors add a brief discussion on this so that the numerous statisticians who work on stratification methods can find this paper and maybe further explore this connection. If I am wrong, and this paper is not doing stratification, the authors should still explain why their method is different.

**Reviewer Concerns:**

I think the concerns of review fKzE that the benchmarks are widely adopoted is missing the point. The choice of hyper-parameters is a valid concern but still the authors address this well in their reply. Even if more experiments are needed to find the hyper-parameters, there is great value to obtain these from proxy models and extrapolate.

**Reviewer Scores:**

I think the method is novel, the problem is real and very significant and the empirical evaluation is very good. Therefore I was not convinced by the negative reviewer scores and I recommend that this paper is accepted.

---

### Decision · Program_Chairs · 2026-01-26

Accept (Poster)